# Voltage-Seq: all-optical postsynaptic connectome-guided single-cell transcriptomics

Veronika Csillag[1], Marianne Hiriart Bizzozzero[1], J. C. Noble ●[2], Björn Reinius ●[2] & János Fuzik ●[1]✉

Understanding the routing of neuronal information requires the functional characterization of connections. Neuronal projections recruit large postsynaptic ensembles with distinct postsynaptic response types (PRTs). PRT is typically probed by low-throughput whole-cell electrophysiology and is not a selection criterion for single-cell RNA-sequencing (scRNA-seq). To overcome these limitations and target neurons based on specific PRTs for soma harvesting and subsequent scRNA-seq, we created Voltage-Seq. We established all-optical voltage imaging and recorded the PRT of 8,347 neurons in the mouse periaqueductal gray (PAG) evoked by the optogenetic activation of ventromedial hypothalamic (VMH) terminals. PRTs were classified and spatially resolved in the entire VMH-PAG connectome. We built an onsite analysis tool named VoltView to navigate soma harvesting towards target PRTs guided by a classifier that used the VMH-PAG connectome database as a reference. We demonstrated Voltage-seq by locating VMH-driven γ-aminobutyric acid-ergic neurons in the PAG, guided solely by the onsite classification in VoltView.

Neuronal information flows through synaptic connections, and modulates large postsynaptic populations in a cell-type-specific manner. Neuronal types can be characterized by morphology, anatomic position, intrinsic excitability, gene expression profile and connectivity. Patch-Seq[1–3] pioneered the molecular characterization of neurons classified by whole-cell patch-clamp[4] recording of intrinsic excitability. To date, synaptic connectivity is probed with whole-cell patch-clamp as it requires the detection of subthreshold postsynaptic potentials (PSPs). The throughput of this technique (-12 neurons per day) is low for the efficient mapping of diverse PRTs in a large postsynaptic population. Finding neurons with specific PRTs for further investigation requires high-throughput connectivity testing and the detection of both subthreshold and suprathreshold membrane potential changes. Genetically encoded fluorescent voltage indicators (GEVIs)[5,6] faithfully report subthreshold voltage changes of both polarities[5] with reliable temporal dynamics and can capture single action potentials (APs).

Voltage imaging has high throughput[7] for simultaneous optical recording of dozens of neurons.

The PAG is a midbrain structure processing panicogenic stimuli[8], and involved in the regulation of autonomic functions[9] and motivated behaviors[10]. PAG receives a strong excitatory input from the VMH[11]. The VMH-PAG axons cover a 2-mm-long anterior–posterior (A–P) range of the dorsal, dorsolateral, and lateral PAG (d, dl, lPAG, respectively). The cell-type- and circuit-motif-specific routing of VMH information in the local PAG circuitry is poorly understood due to its large anatomical extension and high neuronal diversity[12]. We used the VMH-PAG pathway as a model to optimize Voltage-Seq methodology, to all-optical voltage image PRTs, and to select specific neurons for somatic harvesting and subsequent scRNA-seq.

First, we set up all-optical voltage imaging ex vivo implementing the Voltron sensor[7]. Our tiled all-optical imaging has a high throughput of probing up to 1,000–1,500 connections per animal. Next, we

[1]Department of Neuroscience, Karolinska Institute, Stockholm, Sweden. [2]Department of Medical Biochemistry and Biophysics, Karolinska Institute, Stockholm, Sweden. ✉e-mail: janos.fuzik@ki.se

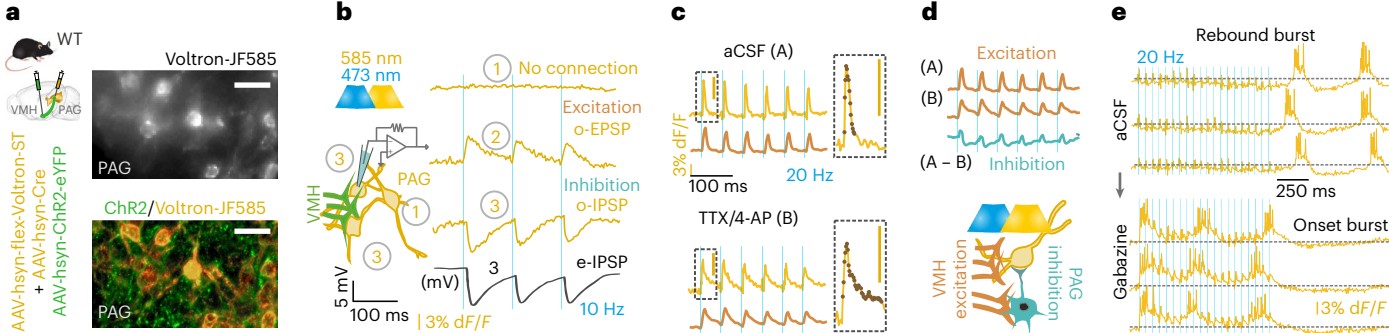

**Fig. 1 | All-optical postsynaptic voltage imaging. a–e**, Voltron traces were reversed, 473 nm was 2.5 mW mm$^{-2}$ and 585 nm power was 14 mW mm$^{-2}$. **a**, Scheme of viral expression of ChR2 in the VMH-PAG pathway and Voltron-ST in the PAG ($N = 14$). Epifluorescent image of neurons with JF-585 signal (white) (top right). Confocal image of the same neurons with ChR2 (green) and JF-585-Voltron-ST (gold) labeling in the PAG (bottom right) (scale bar, 30 μm). **b**, Scheme of simultaneous all-optical voltage imaging and whole-cell patch-clamp recording (left). O-phys traces (average of seven traces, gold) of three neurons (1, 2 and 3) from top to bottom: neuron with no connection (1), neuron with excitation (o-EPSP) (2) and inhibition (o-IPSP) (3), and e-phys trace (average of seven traces, black) of the whole-cell-recorded neuron (3) confirmed inhibitory postsynaptic responses (e-IPSP) (right). **c**, All-optical

compound o-PSPs in aCSF (A) (top, average of seven traces, gold) and in TTX (1 μM)/4-AP (5 mM) (B) (bottom, average of seven traces, gold) with the moving averages below (average of seven traces, brown). Inserts show the kinetics of the compound signal (top) and o-EPSP (bottom) with the detected datapoints overlaid on the o-phys traces. **d**, Top, subtraction of moving averages of excitatory (brown) A and B to extract the disynaptic inhibitory signal component (blue) (A − B) eliminated by TTX/4-AP. Scheme of all-optical voltage imaging of a PAG neuron receiving excitation from VMH and disynaptic inhibition from putative local PAG circuitry (bottom). **e**, All-optical voltage imaging of PAG neuron with inhibition during Op of VMH input and rebound burst firing after the Op (top, 'Rebound burst'); same PAG neuron in bath-applied GABA$_A$ ionotropic receptor (GABAAR) antagonist, Gabazine (10 μM).

generated a whole-structure synaptic connectome of the VMH-PAG projection. Spatial mapping of this connectome revealed the topography of distinct PRTs in the entire PAG. Next, we built an interactive onsite analysis named VoltView, which gave an overview of ~30–80 all-optical imaged PRTs in 1 min. We added a classifier incorporating the generated VMH-PAG connectome data. With that, VoltView could onsite-classify PRTs and navigate a recording- or harvesting pipette to neurons with user-defined target PRTs. We tested Voltage-Seq to locate sparse γ-aminobutyric acid (GABA)ergic neurons in the VMH-PAG guided by the onsite analysis in VoltView. Remarkably, with Voltage-Seq, we identified a VMH-PAG GABAergic feed-forward disinhibitory circuit motif and, using transcriptomics, we identified a neuromodulator that regulates this disinhibitory motif.

## Results

### All-optical postsynaptic voltage imaging
We established and optimized all-optical voltage imaging ex vivo, using the Voltron[7] sensor. We coinjected a virus to express Cre recombinase and another to express Cre-dependent soma-targeting Voltron (Voltron-ST) (Fig. 1a). With this viral combination, we achieved a cell-type-independent Voltron-ST labeling in ~35–40% of PAG neurons (Extended Data Fig. 1a). For all-optical connectivity testing, we also expressed Channelrhodopsin-2 (ChR2) in the VMH (Fig. 1a). We fluorescently labeled Voltron with the Janelia Fluor 585 HaloTag (JF-585) and configured the light path accordingly (Extended Data Fig. 1b). We optimized acquisition speed to 600 Hz, which captured all the optical action potentials (o-APs) in three to four frames (Extended Data Fig. 1c). At this frame rate, a firing rate up to ~125 Hz and changes in o-AP half-width could be detected (Extended Data Fig. 1d–f). With simulated PSPs, we validated the detection limit of ~2–3 mV optical-, excitatory and inhibitory PSPs (o-EPSPs, o-IPSPs) (Extended Data Fig. 1g,h). We found that ~14 mW excitation of JF-585 had a negligible 0.31 ± 0.2 mV (mean ± s.d.) crossactivation of ChR2 (Extended Data Fig. 2a–c,f). ChR2-evoked synaptic release was not influenced by the JF-585 excitation (Extended Data Fig. 2d,e,g). We all-optically imaged both VMH-PAG o-EPSPs and o-IPSPs confirmed by paralleled e-IPSP recording in the imaged PAG neuron (Fig. 1b). All-optical recordings revealed 473 nm light-induced narrow artefacts with reversed polarity to o-APs, which

were removed (Extended Data Fig. 2h). We could detect compound o-EPSP/o-IPSP responses with a narrow profile (Fig. 1c), which displayed o-IPSP upon depolarization (Extended Data Fig. 2i). We used antagonist pharmacology to dissect a compound o-PSP of a putative disynaptic motif (Fig. 1c,d). A rebound-bursting neuron confirmed the codetection of large slow hyperpolarization (300–500 ms), rhythmic depolarization and fast spiking (Fig. 1e). The involvement of GABA was predicted by the compound o-PSPs during optical stimulation (Op) and validated by GABA$_A$ antagonist pharmacology that turned rebound-bursting into onset-bursting (Extended Data Fig. 2j). Substantial bleaching of JF-585 occurred after ~3–4 min of imaging (Extended Data Fig. 2k). Taken together, optimized all-optical imaging could detect the firing activity, bursting, mono- and disynaptic excitatory and inhibitory subthreshold events.

### Classification of all-optical postsynaptic response types
To all-optically image the entire VMH-PAG connectome, we designated the PAG area with high density of VMH axons based on a three-dimensional (3D) axonal map (Extended Data Fig. 3a and Fig. 2a). We all-optically imaged two to three planes in each field of view (FOV), each with 30–80 neurons, tile-covered the PAG with six to seven FOVs (200 × 350 μm$^2$) on five to seven brain slices per mouse in seven mice and imaged 6,911 VMH-PAG neurons (Fig. 2b, and Extended Data Fig. 3b,c). Somatic region of interests (ROIs) were detected by implementing Cellpose segmentation[13]. Optical-physiology (o-phys) traces were extracted (Fig. 2c) and o-AP peaks, subthreshold (o-Sub) kinetics, o-EPSPs, o-IPSPs and burst activity were detected on each o-phys trace (Fig. 2d). Periods of burst were detected based on o-Sub kinetics and validated by the firing frequency during the detected periods (Extended Data Fig. 3c–e). We validated the agreement of bursting PRTs and burster firing type of PAG neurons (Extended Data Fig. 3d,f). We designed Concentric analysis to validate the somatic origin of o-phys traces based on the o-AP peak amplitudes and length of bursts (Extended Data Fig. 3g–i). Overall, ~89% of the imaged PAG neurons were connected to the VMH based on the detection of more than three o-PSPs, 4% of PAG neurons had o-IPSPs. All-optical testing evoked AP firing in ~67% of PAG neurons (Fig. 2e). To classify the VMH-PAG PRTs, we extracted 29 o-phys parameters (Extended Data

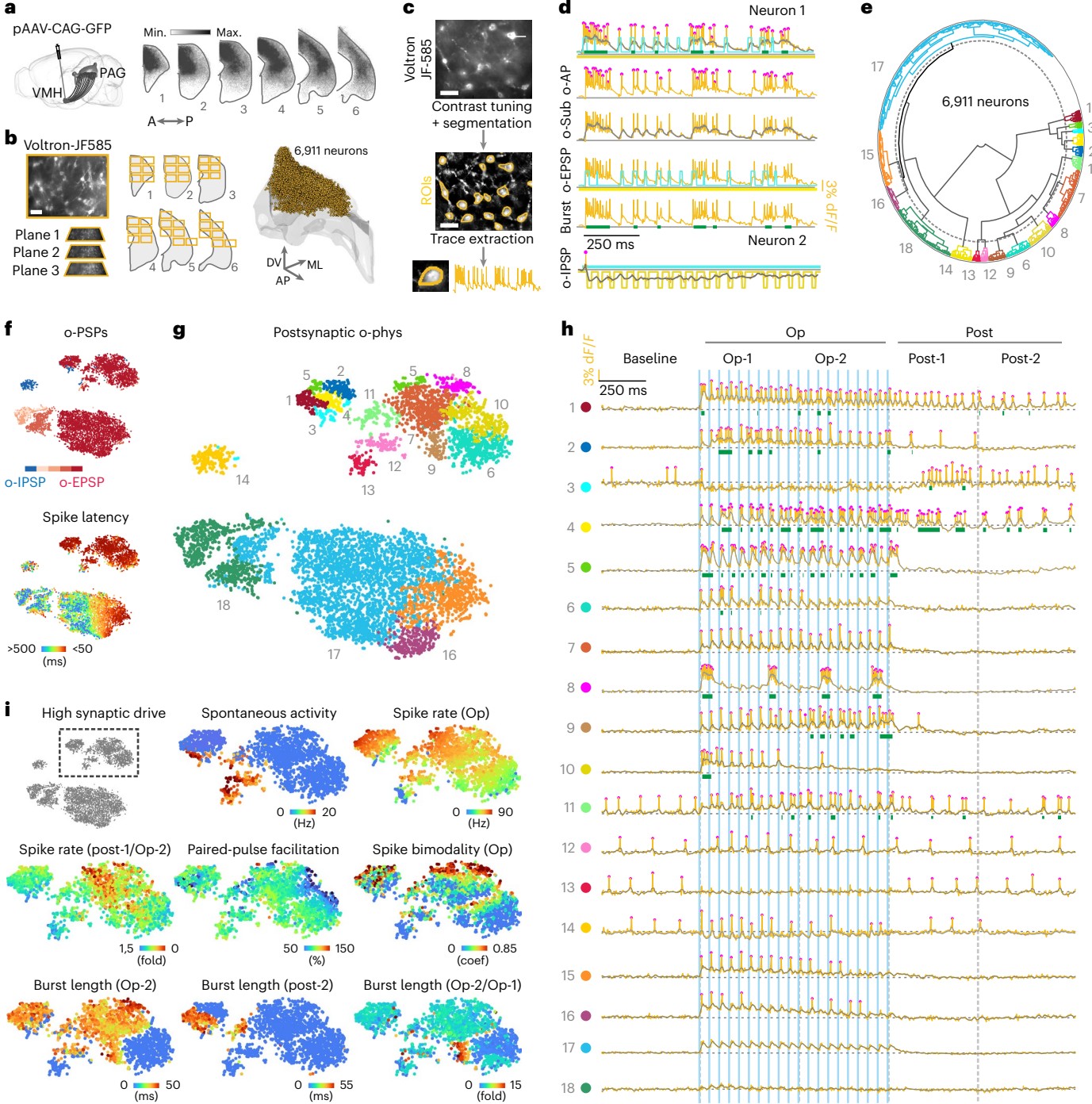

**Fig. 2 | Classification of all-optical postsynaptic response types.**
**a**–**g**, Voltron traces were reversed, 473 nm was 2.5 mW mm$^{-2}$ and 585 nm power was 14 mW mm$^{-2}$. **a**, Scheme of viral GFP labeling of VMH-PAG (left) and coronal bins of our 3D axonal map shows PAG areas with high VMH axonal coverage (right) (N = 2). **b**, Example FOV with JF-585-Voltron-ST PAG neurons and illustration of all-optical tile-imaged PAG slices (left) (scale bar, 50 μm). 3D plot of PAG with the 6,911 neurons of the VMH-PAG all-optical connectome (right, N = 7). **c**, representative FOV frame average of JF-585-Voltron-ST PAG neurons (top) and contrast-tuned frame average with the yellow contours of segmented ROIs (middle), example o-phys trace extracted from an ROI (bottom) (scale bar, 35 μm). **d**, Detection of o-APs, O-Sub, o-EPSPs and o-IPSPs, Burst activity. **e**, Polar dendrogram of agglomerative hierarchical clustering of the VMH-PAG connectome with the identified clusters numbered and colored.

**f**, t-SNE plot of the VMH-PAG connectome color coded by the average number of detected o-IPSPs (blue) or o-EPSPs (red) (shades of red for ranges: 1–5; 5–10; 11–15; 16–20) (top), or color coded by the latency of the first AP during the Op (bottom). **g**, t-SNE plot of the identified o-phys clusters, number and color code is identical to **e**. **h**, Scheme on top details the temporal segments of all-optical sweeps with Op-1, first half of Op; Op-2, second half of Op; Post-1, first half of after-Op; Post-2, second half of after-Op. Representative o-phys traces illustrate PRTs of the identified o-phys clusters (color code and number **e** and **g**) blue bars indicate the 20 Hz 473 nm Op. **i**, t-SNE of the VMH-PAG postsynaptic o-phys (gray), crop of 'High synaptic drive' clusters, which are enlarged in the following t-SNEs to map clustering parameters (abbreviation in brackets for each parameter indicates the temporal segment of the parameter extraction); coef: Sarle's bimodality coefficient.

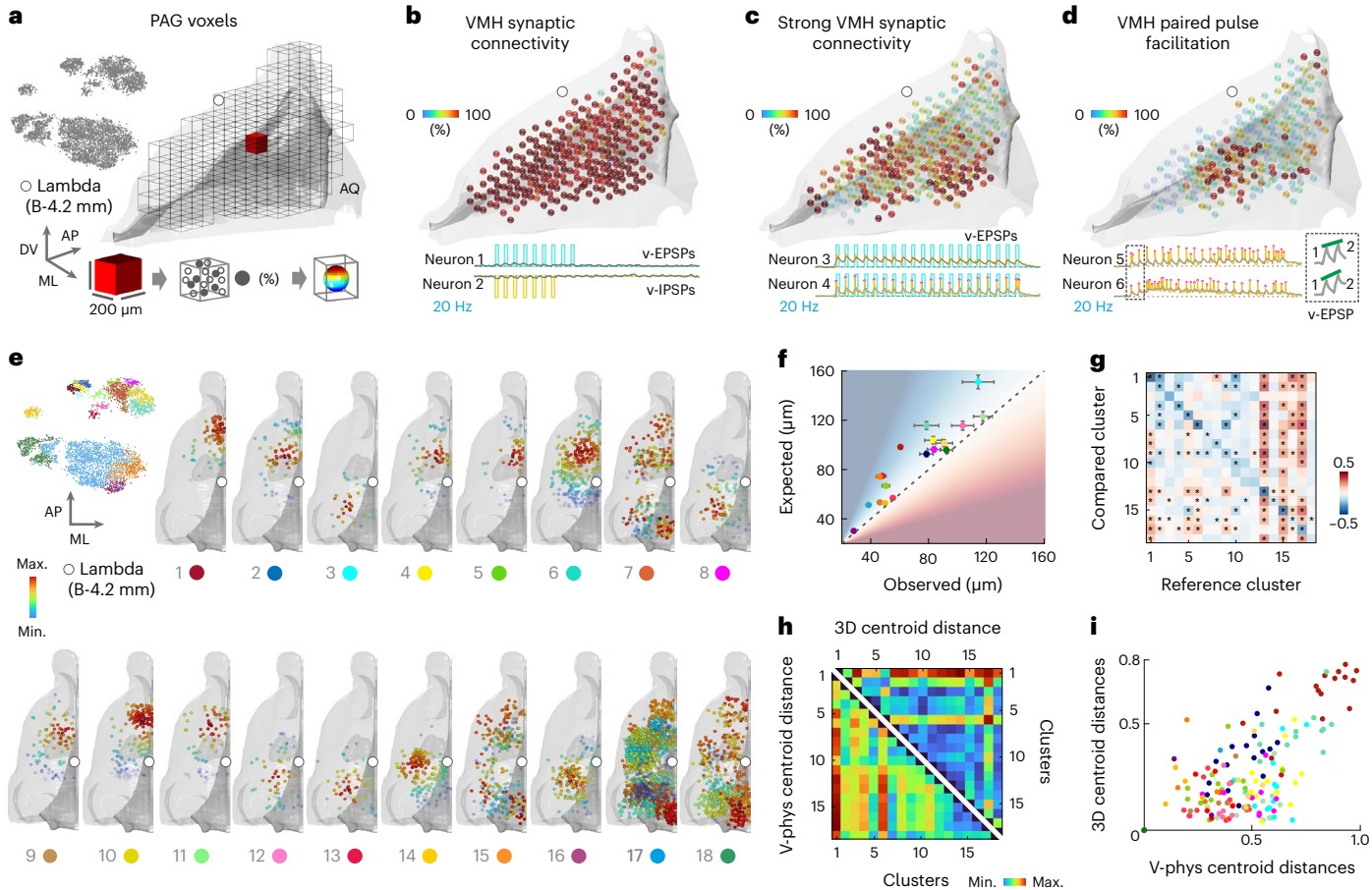

**Fig. 3 | Spatial topography of postsynaptic connectome. a–h**, Voltron traces were reversed. **a**, *t*-SNE of the VMH-PAG connectome in gray, no cluster information was used (top left). 3D PAG model of hemisphere subdivided to voxels (grid) with one voxel in the middle (red). Illustration of a 200 × 200 μm² voxel and the neurons inside the voxel fulfilling a criterion that defined the color code from 0% (blue) to 100% (red), transparency with 0% being invisible (bottom). **b**, 3D PAG model with voxel-mapping of overall VMH connectivity, example PRs fulfilling the low-cut criterion of displaying more than three PSPs (Neurons 1 and 2). **c**, 3D PAG model with voxel-mapping of strong VMH synaptic connectivity, example PRs fulfilling the low-cut criterion of displaying >18 PSPs (Neurons 3 and 4). **d**, 3D PAG model with voxel-mapping PP facilitation, example PRs exerting facilitating PP, insert shows first two o-EPSPs with green line highlighting the facilitation (Neurons 5 and 6). **e**, *t*-SNE of the VMH-PAG

connectome color coded by cluster identity as in Fig. 2e,g,h (top left); 3D PAG model of one hemisphere for each o-phys cluster with the spatial density-core mapping. Colors from red to blue go from highest to lowest spatial density, transparency follows the color code with blue being invisible; in **a–e** white circle indicates the Lambda coordinate (Bregma (B)-4.2 mm). **f**, Observed AMD between neurons of the same cluster versus the AMD expected by chance calculated by repeated shuffles of cluster identity (error bars: ± s.e.m. (bootstrapped), observed (red shade), expected (blue shade)). **g**, AMD of neurons of o-phys clusters was probed across all 18 clusters (*$P < 3 × 10^{-5}$). **h**, Side-by side visualization of o-phys cluster centroid distances and spatial cluster centroid distances (all axes are cluster labels). **i**, Scatter plot of o-phys cluster centroid distances versus spatial cluster centroid distances, colors code cluster identity as in **e**.

Fig. 4a) and performed unbiased hierarchical clustering. We classified the 6,911 PRTs into 18 distinct clusters (Fig. 2f,g and Extended Data Fig. 4b,c) and we identified persistent activity after the Op (cluster 1,4),; rhythmic bursting with 3–4 Hz (cluster 8), separate burst upon each Op (cluster 5); time-locked single o-AP upon each Op (cluster 7); strongly depressing (cluster 10,6) and strongly facilitating short-term synaptic plasticity (cluster 9); facilitating paired-pulse (PP) plasticity (cluster 2,4,9,16); depressing PP plasticity (cluster 15); spontaneous firing activity preceding the Op (cluster 3,11,12,13,14); inhibitory responses (cluster 14); inhibitory responses with rebound firing or bursting (cluster 3); subthreshold response (cluster 17) and weak or not detectable connection (cluster 18) (Fig. 2h). Suprathreshold PRTs have both pre- and postsynaptic components, indicating the interaction of these features. The more synaptic drive is in the PRT, the more the postsynaptic neurons reveal their intrinsic properties (Extended Data Figs. 3f and 4d,e). We validated the lack of animal batch effect across *t*-distributed stochastic neighbor embedding (*t*-SNE) regions of o-phys clusters (Extended Data Fig. 4f). O-phys parameters mapped well across

PRT types displaying suprathreshold responses, referred to as 'High synaptic drive' (Fig. 2i). In summary, our workflow and analysis could resolve the diversity of PRTs with well-defined clusters. The VMH-PAG all-optical connectome revealed the characteristic parameters, the total numbers and proportions of subthreshold and suprathreshold PRTs across the entire PAG (Extended Data Fig. 4g,h).

## Spatial topography of postsynaptic connectome

We post hoc extracted the *XYZ* coordinates of each 6,911 imaged neuron and spatially mapped the VMH-PAG connectome (Extended Data Fig. 5a,b). For spatial PRT-independent overview, the PAG was divided to voxels and, in each, the percentage of PRTs fulfilling a criterion was calculated (Fig. 3a). The voxel-criterion of PRTs with more than three o-PSPs displayed a homogenous distribution of high-percentage voxels, confirming complete coverage of connections throughout the VMH-PAG connectome (Fig. 3b). The voxel-criterion of PRTs with >18 o-PSPs visualized stronger connections in a spatial pattern (Fig. 3c) capturing the more anterior and lateral PAG volumes with higher VMH

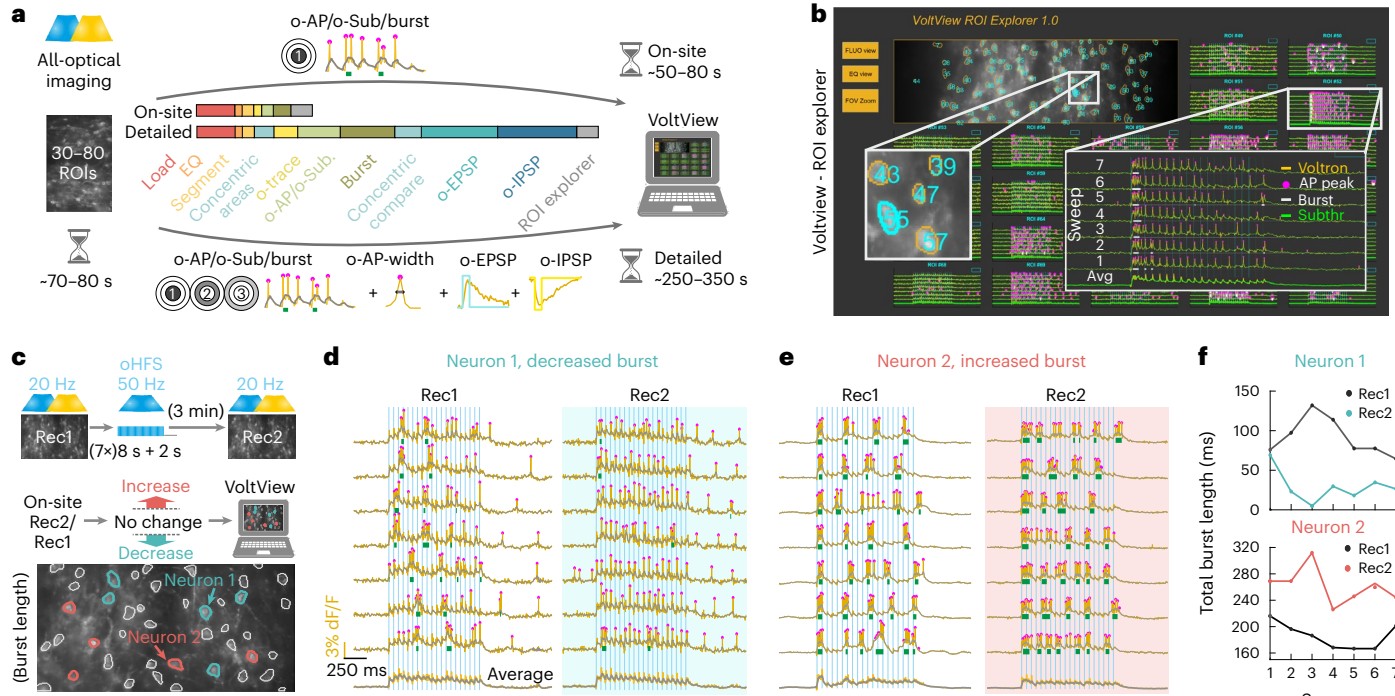

**Fig. 4 | Onsite analysis of all-optical voltage imaging with VoltView.**
**a**–**f**, Voltron traces were reversed, 473 nm was 2.5 mW mm$^{-2}$ and 585 nm power
was 14 mW mm$^{-2}$. **a**, Illustration of the 'On-site' (top arrow) and 'Detailed'
(bottom arrow) analysis of o-phys. Logos summarize the analyzed features
on the arrows. Colored bar segments compare the proportion of time for
different postprocessing, detection and analysis modules (scale bar, 50 μm)
**b**, VoltView graphical user interface with the ROI explorer. Enlarged insert (left)
demonstrates ROI spatial indication with the corresponding number. Enlarged
insert (right) shows the recorded o-phys sweeps with the detected spikes and
bursts. **c**, Rec1 and Rec2 repeated recordings in the same FOV, before and after

50 Hz oHFS; VoltView extracts parameters of the same neurons and compares
them across Rec1 and Rec2. Decreased bursting is indicated with blue ROI
contours, increased bursting is indicated with red ROI contours (bottom)
(scale bar, 30 μm). **d**, Neuron 1, example of bursting PRT with decreased burst
length after oHFS (bottom sweep is the average of the seven sweeps above).
**e**, Neuron 2, example of bursting PRT with increased burst length after oHFS
(last sweep is the average of the seven sweeps above). **f**, Line plots show
comparison of total burst length before (black) and after oHFS (Neuron 1 blue,
top; Neuron 2 red, bottom), dots represent the total burst length in each sweep.

synaptic drive. Voxel-mapping PP facilitation—a qualitative short-term
synaptic plasticity property[14]—highlighted PAG subregions with high
density of facilitating connections (Fig. 3d). Remarkably, mapping
the VMH-PAG connectome by o-phys cluster identity revealed spatial
topography of PRT clusters (Fig. 3e). The coverage of d-dl-lPAG allowed
to inspect the distribution of clusters across these areas (Extended
Data Fig. 5c,d). To test the spatial proximity within and across o-phys
clusters, the average minimal distance (AMD) between neurons of the
same cluster was compared with the chance-level-AMD calculated
by repeated shuffles of cluster identity (Fig. 3f). The observed AMDs
were shorter than chance-level for most clusters, thus o-phys clusters
also formed spatial clusters. The AMD of neurons across the 18 o-phys
clusters was smaller than chance-level between spatially intermingled
clusters 2 and 4, or cluster 9 and 6 (Fig. 3g). The AMD was larger than
chance-level between spatially segregated clusters, such as cluster 8
and 1 (Fig. 3g). The side-by side visualization of o-phys cluster centroid
distances and spatial cluster centroid distances suggested coherence
of the two properties (Fig. 3h). The function of the two distances con-
firmed a pronounced correlation between o-phys cluster identity and
spatial location (Fig. 3i). Taken together, our all-optical connectome
described whole-structure VMH-PAG topography on the quantitative
and qualitative levels using seven mice. This throughput has not been
accessible by any other approach to date.

## Onsite analysis of all-optical voltage imaging by VoltView
To make all-optical experimenting interactive, we developed VoltView.
The 'Detailed' configuration runs thorough analysis to validate signal

source with concentric analysis, extracts o-AP peaks, O-Sub, periods
with burst activity (Burst), o-AP half-width and detects o-EPSPs and
o-IPSPs in a FOV in ~4–5 min (Fig. 4a). The 'On-site' configuration of the
package provides quick access to the basic analyzed features (o-APs,
O-Sub, Burst) of 30–80 neurons in ~1–1.5 min (Fig. 4a). After onsite
analysis, VoltView opens a 'ROI Explorer' for browsing PRTs. The posi-
tion of the currently inspected ROI is indicated in the FOV to spatially
guide the experimenter for further investigation or soma harvesting
(Fig. 4b). Using VoltView, we attempted to identify long-term changes
of PRTs with increased or decreased burst plasticity evoked by a 50 Hz
optogenetic high-frequency stimulation (oHFS) (Fig. 4c). Parameters
of the same neurons were extracted and onsite-compared across imag-
ing sessions (Rec1, Rec2). The comparison had a 25%-change cutoff to
identify ROIs with increased or decreased total burst length (Fig. 4c).
VoltView identified a neuron with robustly decreased bursting where
slow bursting turned into continuous firing after oHFS (Fig. 4d,f).
A neighboring neuron displayed opposite change with increased burst
length where the rhythmicity of bursts blended into a tone, shown
by the sweep average (Fig. 4e,f). Altogether, VoltView analysis made
ex vivo all-optical imaging interactive by onsite selection of neurons
with specific PRT based on user-defined firing property or connection
plasticity features.

## VoltView-guided Voltage-Seq
High-throughput all-optical connectivity testing probes dozens
of connections simultaneously, up to 1,000–1,500 the same day
(Fig. 5a). To onsite-navigate to specific PRTs in such a large datastream,

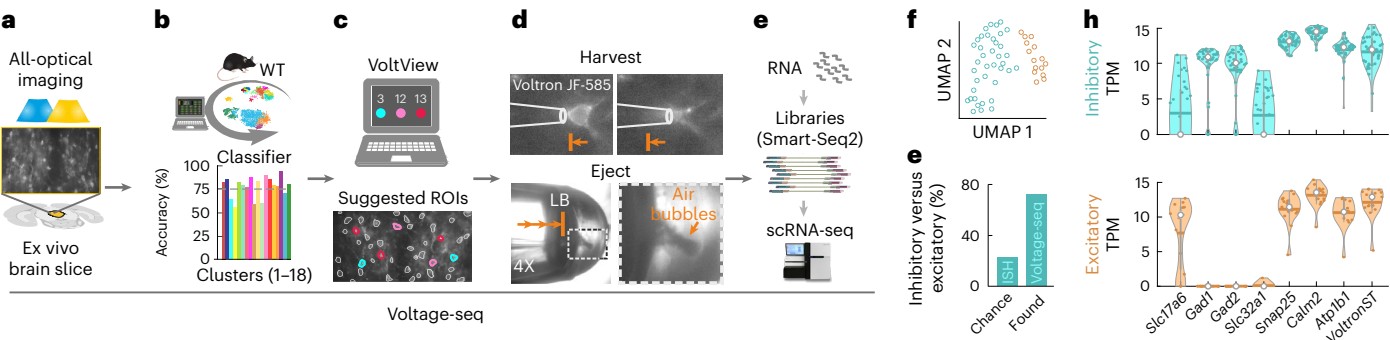

**Fig. 5 | VoltView-guided Voltage-Seq. a–e**, Voltage-Seq workflow. **a**, All-optical voltage imaging; scale bar, 50 µm. **b**, Onsite analysis with classification (classifier efficacy for each cluster on the bar plot color coded by cluster identity). WT, wild type. **c**, VoltView suggests ROIs of a user-defined cluster; scale bar, 50 µm. **d**, Soma harvesting (upper row) (scale bar, 10 µm) in ×20 magnification with ejection (bottom row) under ×4 magnification validated by air bubbles upon ejection of sample in the LB. **e**, cDNA library preparation and scRNA-seq. **f**, UMAP of unbiased clustering of Voltage-Seq RNA-transcriptome (inhibitory neurons, blue; excitatory neurons, brown; *n* = 60 cells, *N* = 3 mice). **g**, Bar plot comparing the proportion of GABAergic versus glutamatergic neurons in our sample

(Found, Voltage-Seq) compared with chance-level (Chance, ISH). **h**, Violin plots show log$_2$ TPM levels of neurotransmitter modality defining *Slc17a6* (Vesicular Glutamate Transporter 2), *Gad1* (67 kDa glutamic acid decarboxylase), *Gad2* (65 kDa glutamic acid decarboxylase), *Slc32a1* (vesicular GABA transporter) and pan neuronal genes as sample controls, *Snap25* (synaptosomal-associated protein 25), *Calm2* (calmodulin 2), *Atp1b1* (sodium/potassium-transporting ATPase subunit beta 1), *Voltron-ST* (soma-targeting Voltron) in the GABAergic (inhibitory, blue) and glutamatergic (excitatory, brown) clusters; horizontal bars represent mean, white dots represent median.

we added a classifier to VoltView with our VMH-PAG connectome data as a reference (Fig. 5b). VoltView suggested neurons classified into user-defined PRT clusters (Fig. 5c) for soma harvesting (Fig. 5d) and for subsequent scRNA-seq (Fig. 5e). We named the workflow Voltage-Seq. To test Voltage-Seq, we attempted to find sparse GABAergic neurons within the VMH-PAG connectome. GABAergic neurons of d-dlPAG had been reported to be depolarized and often spontaneously firing to provide tonic inhibition[15]. Clusters 3, 13 and 12 had 42%, 95% and 100% spontaneous firing (Extended Data Fig. 6a), and were denser in the posterior PAG in our data (Extended Data Fig. 6b), and localized in the d-dlPAG, similarly to the distribution of PAG GABAergic neurons (Extended Data Fig. 6c). To probe the correlation of GABAergic identity and PRTs of cluster 3, 12 and 13, we all-optical imaged 1,436 GABAergic neurons in the posterior PAG of four VGAT-Cre mice (Extended Data Fig. 6d). Cluster-load analysis of VGAT data confirmed large wild-type cluster-load in clusters 3, 12 and 13 (Extended Data Fig. 6e). We set VoltView to suggest neurons of cluster 3, 12 and 13 during all-optical imaging of VMH-PAG PRTs and harvested 60 neurons from three wild-type mice. The harvesting protocol was optimized for Voltage-Seq, for scRNA-seq we used Smart-Seq2 (ref. 16). The detection of ~6,000 genes per cell confirmed high RNA-transcriptome quality (Extended Data Fig. 6f–h). The chance-level of finding GABAergic versus glutamatergic neurons was estimated to be ~22% based on in situ hybridization (ISH) labeling density of GABAergic (*Slc32a1, Gad1, Gad2*) versus glutamatergic (*Slc17a6*) molecular markers (Fig. 5g and Extended Data Fig. 6i,j). Unbiased clustering of Voltage-seq (Fig. 5f) showed three-times higher (72% (39/54)) ratio of GABAergic neurons (Fig. 5g,h), more than expected to find by chance. Taken together, Voltage-Seq successfully identified sparse GABAergic neurons of the VMH-PAG connectome based on the onsite classification in VoltView and could provide access to the molecular identity of the same neurons.

**Neuronal identity and neuromodulation in the Switch motif**
During all-optical voltage imaging in VGAT-Cre mice, we found a circuit phenomenon that had not yet been described in optogenetic experiments before. We observed PRTs we named 'Switch' response, where converging excitatory and inhibitory synaptic inputs dominated in a switching manner. The Switch behavior was stable and reproducible (Extended Data Fig. 7a). Gabazine eliminated the inhibitory synaptic input resulting in only excitatory responses (Extended

Data Fig. 7b). Switch responders were likely to be bistable neurons, which can maintain prolonged depolarization, long overreaching the excitatory stimulus[17–19]. Our Voltage-Seq dataset also contained Switch responders, as they are often firing at the baseline during switching. We found that they were *Gad1*⁺/*Gad2*⁺ GABAergic neurons (Extended Data Fig. 8a). To unveil neuronal identity in the Switch motif, we all-optically imaged VMH-PAG in a wild-type mouse and browsed the PRTs in VoltView to find Switch responses (Fig. 6a). After locating a Switch response (Fig. 6b), we whole-cell-recorded the post-synaptic neurons. We characterized the intrinsic excitability and found both burst firing (6/8) and regular spiking (2/8) types (Fig. 6c). Thus, Switch motif was not rendered to only one specific neuronal type but is a circuit domain integrating multiple neuronal types. To confirm the disynaptic nature of inhibition we measured (mean ± s.d.) 1.52 ± 0.2 ms EPSC and 4.86 ± 2.36 ms IPSC delay in burster and 1.86 ± 0.73 ms EPSC and 4.72 ± 1.78 ms IPSC delay in regular spiking neurons in voltage-clamp mode during Op (Fig. 6d). Furthermore, we probed GABAergic identity with *Slc32a1* ISH and found that all the investigated neurons with Switch response were GABAergic, in agreement with our Voltage-seq transcriptome data (Fig. 6e). Differential expression (DE) analysis of excitatory and inhibitory clusters highlighted putative marker genes of GABAergic neurons (for example, *Nrxn3, Pnoc, Gata3*) (Fig. 6f and Extended Data Fig. 8c). Based on our Voltage-seq RNA-transcriptome, amongst other GABAergic cells, the neurons with Switch response were expressing *Tacr1* and *Tacr3* encoding neurokinin-1 and -3 (NK-1 and NK-3) receptors (Fig. 6f and Extended Data Fig. 8b). To probe neuromodulation of the Switch motif mediated by NK-1 and NK-3, we bath-applied Substance P (SP)—the endogenous ligand of the receptors[20]. Multiple video comparisons in VoltView showed increased PRT firing frequency only in a subset of neurons (5/58) (Fig. 6i). In a strong Switch responder, we could voltage image SP neuromodulation (Fig. 6g,h). Subthreshold excitatory responses turned suprathreshold, while inhibitory responses were suppressed and rebound bursts were eliminated. This could be due to the SP-induced inward Na⁺-current increasing the firing probability and counteracting inhibition[21]. Furthermore, we located this neuron in VoltView and, with whole-cell recording, we identified a burst firing type, which could exert rebound burst firing (Fig. 6j). We validated the GABAergic identity of the same neuron with *Slc32a1* ISH (Fig. 6k). In summary, our Voltage-seq methodology could allow closer

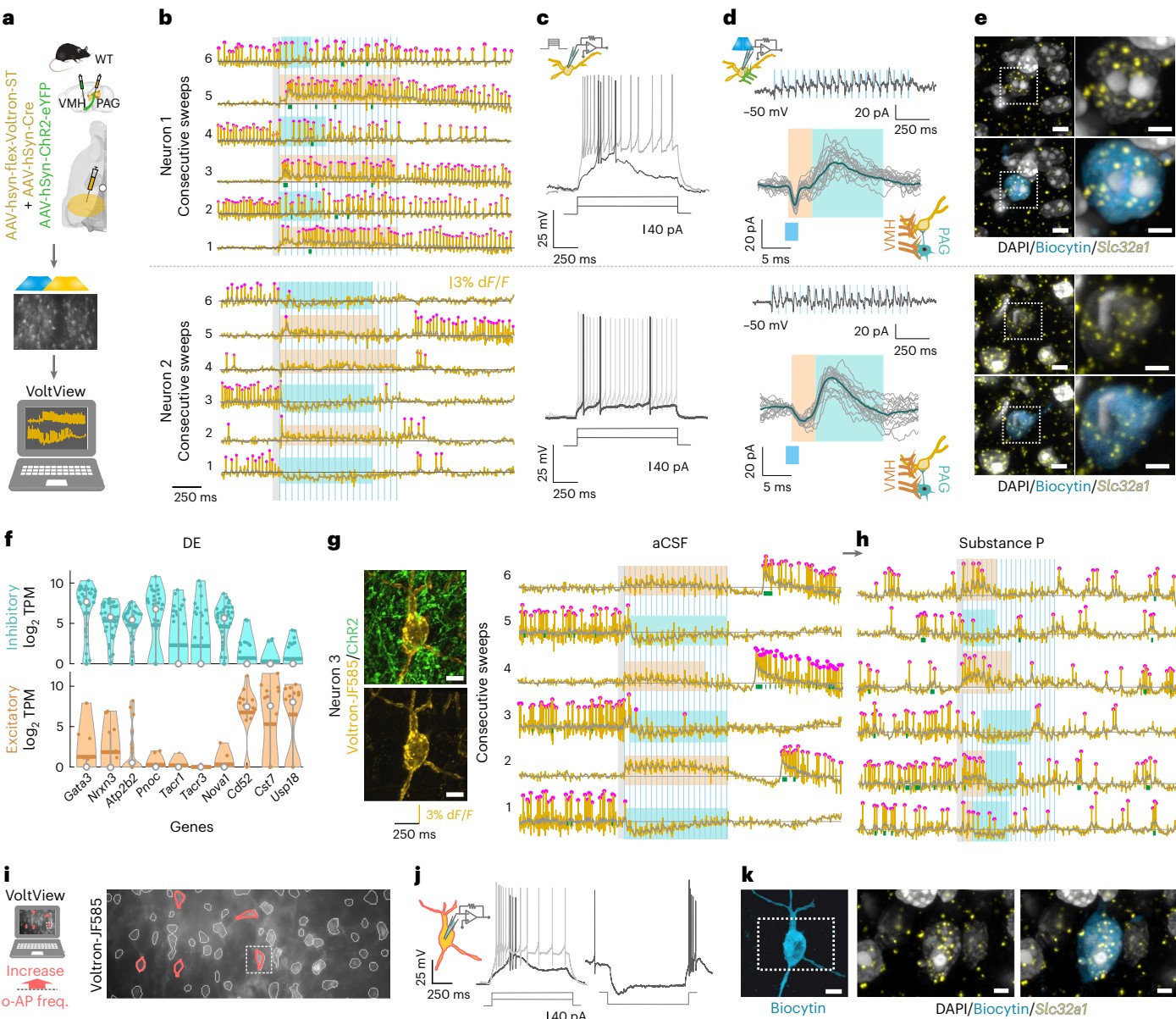

**Fig. 6 | Neuronal identity and neuromodulation in the Switch motif.**
**a–k**, Voltron traces were reversed. Power 2.5 mW mm⁻² for 473 nm and 14 mW mm⁻² for 585 nm. **a**, Scheme of experiment with expression of ChR2 and Voltron-ST in the VMH-PAG (top), all-optical voltage imaging (middle) and onsite analysis in VoltView (bottom) (scale bar, 40 μm). **b**, Switch responder Neuron 1 (top), Neuron 2 (bottom), (cyan rectangles indicate inhibition, brown rectangles indicate excitation). **c**, Current-clamp identified bursting type of Neuron 1 (top) and regular spiking type of Neuron 2 (bottom). **d**, Neurons 1 and 2: voltage-clamp trace of responses upon 20 Hz Op of VMH terminals (−50 mV, average of seven sweeps). Overlaid responses with e-EPSCs (brown rectangle) and e-IPSCs (cyan rectangle) **e**, Neurons 1 and 2: 4,6-diamidino-2-phenylindole (white), *Slc32a1* (yellow), Biocytin (blue). Inset, higher magnification (scale bars, 5 μm, insets 2 μm). **f**, Violin plot of DE genes of GABAergic (cyan) and glutamatergic (brown) Voltage-Seq neurons (white circle, median; horizontal bar, mean). *Gata3*

(GATA Binding Protein 3), *Nrxn3* (Neurexin 3), *Atp2b2* (ATPase Plasma Membrane Ca²⁺ Transporter 2), *Pnoc* (Pre-nociceptin), *Tacr1* (Tachykinin Receptor 1), *Tacr3* (Tachykinin Receptor 3), *Nova1* (NOVA Alternative Splicing Regulator 1), CD52 (Campath-1 antigen), *Cst7* (Cystatin F) and *Usp18* (Ubiquitin specific peptidase 18). **g**, Voltron-JF-585-labeled Neuron 3 (gold) and ChR2-expressing axons (green) (left) (scale bar, 6 μm). Consecutive o-phys traces with a strong Switch response in aCSF. **h**, Consecutive o-phys traces of Neuron3 after bath-application of SP (1 μM) (cyan rectangles, inhibition; brown rectangles, excitation in **g** and **h**). **i**, Multiple video comparison in VoltView indicated neurons with increased AP firing upon SP wash-on (red ROIs) (insert marks Neuron3) (scale bars, 50 μm). **j**, Whole-cell recording of Neuron3 revealed the burst firing type of the Switch responder (middle), with rebound burst firing after negative current injection (right). **k**, Biocytin-filled (blue) Neuron3. Inset, 4,6-diamidino-2-phenylindole (white), *Slc32a1* (yellow) and Biocytin (blue), (scale bars, 6 μm, insets 2 μm).

investigation of a disynaptically disinhibitory circuit motif. Switch responders would have been extremely challenging to characterize by whole-cell patch-clamp as they are relatively sparse. Based on the Voltage-seq transcriptomics, we could identify SP as a neuromodulator of the Switch motif, and all-optical image the SP neuromodulation of the Switch motif.

## Discussion

We optimized Voltage-Seq, which combines all-optical-physiology, spatial mapping, onsite classification and RNA transcriptomics to increase the throughput of synaptic connectivity testing and targeted molecular classification of postsynaptic neurons. The literature of VMH-PAG synaptic physiology is, as yet, exiguous. Our approach provided a

detailed insight into the VMH-PAG synaptic connectome with measuring thousands of connections on the qualitative and quantitative levels using only a few animals. Such connectome data in any brain region is a potent starting point for addressing questions concerning PRTs and the adjacent neuronal types.

In functional connectivity studies calcium imaging is suitable to monitor robust suprathreshold activity on multiple neurons simultaneously. However, even the latest GCaMP calcium indicators have low temporal resolution to resolve single APs and low sensitivity to report subthreshold activity, especially inhibition. GEVI imaging gives access to subthreshold events of both polarities and has the temporal resolution necessary to interrogate complex PRTs. All-optical voltage imaging with another opsin/GEVI pair (ChR2/QuasAr6) had been already used in vivo as well and could be used in future applications of Voltage-Seq in vivo.

Comprehensive understanding of the identity of postsynaptic ensembles was so far occluded by the throughput of connectivity probing techniques. Besides, patch-clamp may introduce perturbation of the intracellular ionic milieu; for example, we could not induce the switching behavior of a Switch responder during whole-cell recording. In contrast, Voltage-Seq of circuit motifs could capture the native circuit behavior of the integrated neurons. The interplay of long-range synaptic inputs, local connectivity and intrinsic properties of postsynaptic neurons could be observed and Voltage-Seq could resolve the molecular identity of these neurons.

Voltage-Seq methodology should be more accessible to a wider range of neuroscientists as it needs less hands-on skills compared with patch-clamp-electrophysiology. Voltage-Seq united the power of high-throughput, high-signal-resolution connectivity imaging, interactive analysis and molecular profiling of the imaged neurons, which is a notable advancement in postsynaptic circuit dissection.

## Online content

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

## Methods

### Animals

Experiments were conducted using adult male and female mice, wild-type C57BL/6J (Charles River Laboratories) or the transgenic mouse line VGAT-Cre: *B6J.129S6(FVB)-Slc32a1^{tm2(cre)Lowl}/MwarJ*, Jackson stock no. 028862. All transgenic mice used in experiments were heterozygous for the transgenes. Mice were group housed, up to five per cage, in a temperature- (23 °C) and humidity- (55%) controlled environment in standard cages on a 12/12 h light/dark cycle with ad libitum access to food and water. All procedures were approved and performed in accordance and compliance with the guidelines of the Stockholm Municipal Committee (approval no. N166/15 and 7362-2019).

**Animal cohorts.** C57BL/6J: pAAV-CAG-GFP

$N$ = 2 males: axonal anatomy and histology.
C57BL/6J: pAAV-hSyn-hChR2(H134R)-EYFP
$N$ = 8 males: patch-clamp electrophysiology of technical controls of all-optical crosstalk.
C57BL/6J: pENN-AAV-hSyn-Cre-WPRE-hGH; pAAV-hsyn-flex-Voltron-ST; pAAV-hSyn-hChR2(H134R)-EYFP
$N$ = 7 males: voltage imaging−VMH-PAG connectome 6,911 neurons.
$N$ = 3 males: voltage imaging−Voltage-Seq harvesting of putative GABAergic PAG neurons.
$N$ = 4 males: voltage imaging−finding Switch responders for whole-cell patch-clamp and ISH histology.
VGAT-Cre: pAAV-hsyn-flex-Voltron-ST; pAAV-hSyn-hChR2(H134R)-EYFP
$N$ = 4 males: exclusive imaging of GABAergic PAG neurons.

### Viral constructs

All purified and concentrated adeno-associated viruses (AAV) were purchased from Addgene.

**Anatomy and histology.** pAAV-CAG-GFP (AAV5); Addgene, catalog no. 37825-AAV5; (at titer ≥7 × 10^{-12} viral genomes ml^{-1})

**Voltage imaging.** pAAV-hsyn-flex-Voltron-ST (AAV1); Addgene, catalog no. 119036-AAV1; (at titer ≥2 × 10^{-12} viral genomes ml^{-1})
pAAV-hSyn-hChR2(H134R)-EYFP (AAV5); Addgene, catalog no. 26973-AAV5; (at titer ≥7 × 10^{-12} viral genomes ml^{-1})
pENN-AAV-hSyn-Cre-WPRE-hGH (AAV1); Addgene, catalog no. 05553-AAV1; (at titer ≥1 × 10^{-13} viral genomes ml^{-1})

### Viral injections

**General procedure.** Mice were anesthetized with isoflurane (2%) and placed into a stereotaxic frame (Harvard Apparatus). Before the first incision, the analgesic Buprenorphine (0.1 mg kg^{-1}) and local analgesic Xylocain/Lidocain (4 mg kg^{-1}) was administered subcutaneously. The body temperature of the mice was maintained at 36 °C with a feedback-controlled heating pad. For viral injections a micropipette attached on a Quintessential Stereotaxic Injector (Stoelting) was used. Injections were done with a speed of 50 nl min^{-1}. The injection pipette was held in place for 5 min after the injection before being slowly (100 μm s^{-1}) retracted from the brain. The analgesics Carprofen (5 mg kg^{-1}) was given at the end of the surgery, followed by a second dose 18–24 h after surgery.

**Labeling strategies.** For anatomical characterization and electrophysiological recordings of the VMH-PAG pathway in C57BL/6J mice, 0.3 μl pAAV-CAG-GFP or pAAV-hSyn-hChR2(H134R)-EYFP was unilaterally injected into the VMH (coordinates: anteroposterior (A–P) −1.45 mm, mediolateral (ML) 0.25 mm, dorsoventral (DV) −5.25 mm). Targeting of the PAG was achieved by one (coordinates: A–P −4.1 mm, ML 0.2 mm, DV −1.6 mm) or two (coordinates: A–P −3.9 mm and A–P −4.3 mm, ML 0.2 mm, DV −1.6 mm) unilateral injections of 0.3 μl of a 1:1 mixture of pENN-AAV-hSyn-Cre-WPRE-hGH and pAAV-hsyn-flex-Voltron-ST. We injected the pAAV-hSyn-hChR2(H134R)-EYFP to the VMH and the pAAV-hsyn-flex-Voltron-ST virus to the PAG during the same transcranial surgery.

### Histology

**General procedure.** Mice were deeply anaesthetized with Na-pentobarbital (60 mg kg^{-1}) and transcardially perfused with 0.1 M PBS followed by 4% paraformaldehyde (PFA) in PBS 0.1 M. Brains were removed and postfixed in 4% PFA in PBS 0.1 M overnight at 4 °C and then washed and stored in 0.1 M PBS. Coronal, 50 μm slices were cut using a vibratome (Leica VT1200S). The sections were washed in 0.1 M phosphate-buffer (PB) and mounted on glass slides (Superfrost Plus, Thermo Scientific) and coverslip-covered (Thermo Scientific) using glycerol: 1× PBS (50:50).

**Histology of biocytin-filled neurons.** Brain slices (250 μm thick) containing biocytin-filled neurons and voltron-JF-585 labeling were postfixed in 4% PFA in PB, 0.1 M, pH 7.8) at 4 °C overnight. Slices were repeatedly washed in PB and cleared using CUBIC protocol[22]. First 'CUBIC reagent 1' was used (25 wt% urea, 25 wt% *N*,*N*,*N'*,*N'*-tetrakis (2-hydroxypropyl) ethylenediamine and 15 wt% polyethylene glycol mono-p-isooctylphenyl ether/Triton X-100) for 1 day at 4 °C. After repeated washes in PB, biocytin was visualized using Alexa Fluor 633-conjugated streptavidin (Thermo Fisher, S21375, 1:1,000) at rom temperature for 3 h. For NeuN staining, primary antibody (Millipore, MAB377, 1:1,000, Mouse, IgG1) was incubated overnight at 4 °C and, after repeated washing with PB, second antibody (Jackson Cy5 AffiniPure Donkey Anti-Mouse IgG (H+L), Code: 715-175-151, 1:500) was incubated for 3 h at room temperature. Slices were then rewashed in PB and submerged in 'CUBIC reagent 2' (50 wt% sucrose, 25 wt% urea, 10 wt% 2,20,20'-nitrilotriethanol and 0.1% v/v% Triton X-100) for further clearing. Slices were mounted on Superfrost glass (Thermo Scientific) using CUBIC2 solution and covered with 1.5 mm cover glasses.

**In situ hybridization.** We used RNAscope Fluorescent Multiplex Assay v.2 (catalog no. 323110) to visualize *Slc32a1* in biocytin-filled, voltage-imaged neurons. Brain slices after all-optical voltage imaging were fixed overnight at 4 °C in 4% PFA; on the next day, slices were repeatedly washed in PB. The fluorescence ISH protocol followed the manufacturer's instructions with modified incubation time as slices were 250 μm thick. We incubated the free-floating slices with the ISH probe overnight instead of 2 h on a slide, at 40 °C. On the following day, the sections were washed in wash buffer and treated with AMP-1FL for 30 min at 40 °C and Amp-2FL for 15 min at 40 °C. JF-585 labeling/ fluorescence was almost eliminated by the ISH protocol; thus, we also took images before and after ISH. We attempted to reincubate PFA-fixed tissue in JF-585 HaloTag-dye but it did not relabel Voltron-ST-expressing neurons. Immunostaining of biocytin worked both before and after ISH using the same protocol as above (Histology of biocytin-filled neurons) without the CUBIC clearing steps.

**Confocal imaging.** All confocal images were taken using a Zeiss 880 confocal microscope. CUBIC cleared sections after slice electrophysiology and biocytin or NeuN staining were acquired as z-stacks using a Plan-Apochromat ×20/0.8 M27 objective (imaging settings: frame size 1,024 × 1,024, pinhole 1 AU (Airy unit), Bit depth 16-bit, speed 6, averaging 4). For viral expression overview of coronal cut VMH or PAG, sections were acquired with a Plan-Apochromat ×20/0.8 M27 objective (imaging settings: frame size 1,024 × 1,024, pinhole 1 AU, Bit depth 16-bit, speed 7, averaging 2). For ISH images oil-immersion ×63/1.0 objective was used (imaging settings: frame size 1,024 × 1,024, pinhole 1 AU, Bit depth 16-bit, speed 6, averaging 4). Processing of images was

done in either ImageJ (National Institutes of Health (NIH)) or Imaris v.7.4.2 (Oxford Instruments).

## Brain slice preparation ex vivo

First, mice were anesthetized with intraperitoneal injection of 50 μl Na-pentobarbital (60 mg kg$^{-1}$) and transcardially perfused with 4–8 °C cutting solution, containing 40 mM NaCl, 2.5 mM KCl, 1.25 mM NaH$_2$PO$_4$, 26 mM NaHCO$_3$, 20 mM glucose, 37.5 mM sucrose, 20 mM HEPES, 46.5 mM NMDG, 46.5 mM HCl, 1 mM L-ascorbic acid, 0.5 mM CaCl$_2$ and 5 mM MgCl$_2$. Next, brain was carefully removed and 250 μm thick coronal slices were cut with a vibratome (VT1200S, Leica) in the same 4–8 °C cutting solution. Next, slices were incubated in cutting solution at 34 °C for 13 min, and kept until recording at room temperature in artificial cerebrospinal fluid (aCSF) solution containing 124 mM NaCl, 2.5 mM KCl, 1.25 mM NaH$_2$PO$_4$, 26 mM NaHCO$_3$, 20 mM glucose, 2 mM CaCl$_2$ and 1 mM MgCl$_2$. For Voltron imaging, slices were incubated at room temperature in JF-585 HaloTag (JF-dyes, Janelia) ligands. JF-dyes were dissolved in DMSO to a stock of 1 μM and further diluted to 50 nM in aCSF before use. All solutions were oxygenated with carbogen (95% O$_2$, 5% CO$_2$). All constituents were from Sigma-Aldrich.

## Patch-clamp electrophysiology

C57BL/6J mice were injected with pAAV-hSyn-hChR2(H134R)-EYFP at 10–11 weeks and recorded 12–13 weeks of age (for details on the virus injection, see Viral injections). For patch-clamp recordings, brain slices were superfused with 33–34 °C aCSF at a rate of 4–6 ml min$^{-1}$. Neurons were visualized using a ×60 water-immersed objective (Olympus) in a differential interference contrast (DIC) microscope on an Olympus BX51WI microscope (Olympus). Patch borosilicate (Hilgenberg) pipettes, 7–10 MΩ pulled using a horizontal puller (P-87 Sutter Instruments) were filled with K-gluconate-internal solution containing 130 mM K-gluconate, 5 mM KCl, 10 mM HEPES, 10 mM Na$_2$-phosphocreatine, 4 mM ATP-Mg, 0.3 mM GTP-Na, 8 biocytin, 0.5 EGTA, (pH 7.2 set with KOH). The same intracellular solution was used for both current-clamp and voltage-clamp recordings. Signals were recorded in pClamp v.10.4 (Molecular Devices) with an Axon Multi-Clamp 700B amplifier and digitized at 20 kHz with an Axon Digidata 1550B digitizer (Molecular Devices). Pipette capacitance was compensated, liquid junction potential was not corrected. Neurons recorded in current-clamp mode were held at a membrane potential of −60 mV. 'AP threshold' (mV) was defined as the voltage point where the upstroke's slope trajectory first reached 10 mV ms$^{-1}$. 'AP half-width' (ms) was measured at half the maximal amplitude of the AP. To assess the firing types, the neurons were held at a membrane potential of −70 mV and 1-s-long positive current was injected. To test the ability of neurons to rebound burst, neurons were held at −60 mV and a 1-s-long negative current step was injected. The synaptic properties of VMH projection onto PAG neurons were tested in both voltage- and current-clamp mode on different holding potentials (−70, −60, −50 mV) using multiple light pulse train protocols with 2 ms blue light pulses with ~2.5 mW light power from a Spectra X (Lumencor) LED light source. In some experiments, we bath-applied tetrodotoxin (TTX; 1 μM; Tocris) and 4-AP (5 mM; Sigma-Aldrich) to isolate monosynaptic responses. Inhibitory currents in some cases were blocked pharmacologically by bath-application of GABA$_A$ antagonist Gabazine (10 μM; Sigma-Aldrich). For pharmacological testing of SP effect, SP (1 μM, Tocris) was bath-applied in the aCSF. All parameters were analyzed by procedures custom-written in MATLAB (MathWorks).

## Voltage imaging

After 4–5 weeks of Voltron-ST expression, mice were sacrificed and ex vivo brain slices of 250 μm were prepared. After ~30 min of incubation in 50 nM JF-585 dye dissolved in aCSF, slices were transferred to the recording chamber of the electrophysiology setup. For the imaging we used a digital sCMOS camera (Orca Fusion-BT, Hamamatsu), and

HCImage Live v.4.6.0 (Hamamatsu) frame triggers were sent to the camera with a Arduino Micro microcontroller (Arduino Uno) with 600 Hz. For all-optical imaging we used a dual-band excitation filter (ZET488/594, Chroma) to excite the JF-585 and deliver a 473 nm light for optogenetic stimulation. The 585 nm light excitation intensity was ~14 mW mm$^{-2}$ and 473 nm light intensity was ~2.5 mW mm$^{-2}$ at the slice plane and was delivered by Spectra X (Lumencore) LED light source. JF-585 fluorescent emission was collected with a 20 × 1.0 NA water immersion objective (XLUMPLFLN20XW Plan Fluorit, Olympus). Emitted light was separated from the excitation light with a band-pass emission filter (ET645/75, Chroma) and with a dichroic mirror (T612lprx, Chroma). Magnification was decreased with U-ECA magnification changer to ×0.5. To acquire videos, we used the free LiveImage software triggered by the Arduino to synchronize acquisition with frame triggers.

## Voltage-Seq

**Neuronal soma harvesting.** After voltage imaging of each FOV and consequent onsite analysis in VoltView, we approached the suggested somas one-by-one with a harvesting pipette (1.8–2.5 MΩ) containing 90 mM KCl and 20 mM MgCl$_2$. The entire soma of the selected neuron was aspirated into the pipette within a few seconds by applying mild negative pressure (−50 mPa) measured with a manometer. Switching from DIC to the fluorescent optics to visualize the Voltron-expressing neurons during and after aspiration could confirm the successful harvesting process. Next, the harvesting pipette was pulled out of the recording chamber and then, with the micromanipulator, carefully navigated over and inside a 0.2 ml tube under visual guidance observed by a ×4 Olympus air objective (Olympus). Applying positive pressure, the harvested neuron (~0.5 μl) was ejected into a 4 μl drop of lysis buffer (LB) consisting of 0.15% Triton X-100 (Sigma), 1 U μl$^{-1}$ TaKaRa RNase inhibitor, 1.5 mass U μl$^{-1}$ SEQURNA thermostable RNase inhibitor (catalog no. SQ00201), 2.5 mM dNTP, 17.5 mM dithiothreitol and oligo dT primer (2.5 μM) preplaced in the very tip of the 0.2 ml tight-lock tube (TubeOne). Through the ×4 air objective we could observe the line of small air bubbles coming out of the harvesting pipette tip into the LB as confirmation of the completed ejection. The resultant sample (~4.5 μl) was spun down (15–20 s), placed on dry ice, stored at −80 °C and later subjected to intube reverse transcription.

**Single-cell RNA-sequencing.** Smart-Seq2 (SS2) libraries were prepared as previously described except for the following being changed: instead of recombinant inhibitor, LB contained 1.5 mass U μl$^{-1}$ SEQURNA thermostable RNase inhibitor (as stated above); no additional RNase inhibitor was added to the reverse transcriptase mix; first strand cDNA from harvested neurons was amplified for 22 cycles, cDNA 1 ng was tagmented and amplified with custom 10 bp indexes. Libraries were sequenced using a 150 cycle Nextseq 550 kit (paired end, 74 bp reads).

**Sequencing QC.** Reads were aligned with the mm10 genome using zUMIs (v.2.9.5) using STAR (v.2.7.2a) with transcript annotations from GENCODE (v.M25). Quality control reports for genes detected was performed with intron+exon alignments from zUMIs. Qualimaps was used to report the total reads aligned to intronic and exonic regions.

**DE and clustering.** All expression analyses were performed with exonic reads only. The package Seurat in R was used to perform clustering analyses. Briefly, Smart-seq2 data was normalized to account for sequencing depth (gene count divided by total counts, multiplied by a scale factor of 10,000 and log normalized) and the 1,000 top variable features were found. Expression for each gene was scaled around 0 and a linear dimensionality reduction (principal component analysis) was performed. For the clustering analysis, the K-nearest neighbor graph was constructed with the first five principal components and a Louvian algorithm iteratively grouped cells together using a resolution of 0.75

to obtain two clusters. Markers for each cluster were found using the Wilcoxon rank sum test. The heatmap for UMAP clustered cells and scaled marker expression was generated with ComplexHeatmap.

## Anatomical 3D mapping

Spatial mapping of imaged neurons was done using the common coordinate framework 4 (CCF4) (ref. 23). $X$ and $Y$ coordinates were extracted from the videos relative to the top left corner and when $Z$ coordinates were identified. The videos were taken with a predefined tile positioning. At each A–P coordinate we positioned the top left corner of the video on the most dorsal point of the midline in the right hemisphere. The $XY$ table was calibrated so that it moved on the $X$ and $Y$ axis by the length and width of videos to capture nonoverlapping FOVs. On a hemisphere, we used two (ML) by three (DV) FOVs, posterior from Bregma-4.4 we used three (ML) by four (DV) FOVs as shown in Fig. 2b. The FOVs were mapped using reference lines to match the step of the $XY$ table in every copronal cross-section of the PAG. We generated these reference lines with SHARP-track (Extended Data Fig. 5a). The package was used originally to track electrodes, but it is suitable for constructing reference points or lines in 3D and extract the coordinates to support 3D mapping. Our reference lines went all along the A–P axis of PAG on the most dorsal, lateral and ventral edges, and within the PAG they were spaced by the distance of the size of FOVs recorded. FOV positions were registered during data acquisition and analyzed offline to calculate $X$, $Y$ and $Z$ coordinates of each imaged neuron.

**Spatial core-density mapping.** For each o-phys cluster, for each neuron, the number of neighbors in a 300 μm radius was calculated. The density core of a cluster was defined by the neuron with the highest number of neighbors. The spatial distance of the $X$, $Y$ and $Z$ coordinates of each neuron of the cluster was calculated from the $X$, $Y$ and $Z$ coordinates of the density core neuron. On the 3D core-density maps, neurons were color coded by the distance from the density core starting in red for closest towards blue with highest distance, transparency followed the color code with blue being invisible.

**Axonal mapping.** To label the VMH-PAG axons, we expressed GFP with injection of pAAV-CAG-GFP into the VMH. Coronal cut 50-μm-thick PAG sections of the PFA-fixed brains were mapped manually to an AP coordinate Z, and X,Y coordinates were adjusted by the edges of the PAG. We imaged every third section to cover the PAG with 150 μm A–P intervals, as the overall axonal coverage did not vary drastically in every 50–100 μm interval; this provided an estimate to designate the areas to an all-optical image. For each fluorescent image, a pixel (axonal fluorescent labeling) was segmented out in ImageJ. We transformed the confocal axon images into binary images where every pixel above background-level signal was 1, and every pixel with background-level signal or below was 0. For mapping coronal slices of axonal masks, we used the reference lines generated with SHARP-track and fitted the slices in the 3D PAG model with a custom-written script in MATLAB. We mapped these binary images to our 3D map and calculated the axonal density in 3D (Extended Data Fig. 3a). The 3D map was coronally resliced to six thicker blocks of ~600 μm each.

## Data analysis

**VoltView analysis.** VoltView analyzes all-optical Voltron imaging by default, it is built in such a way that it looks for the 'Blue spikes' delivered by the blue optogenetic stimulation and reported by the FRET Voltron sensor. Also, VoltView has a built-in classifier that can be updated even after each experiment, so that the classification of PRTs will become more and more accurate by involving more and more o-phys data in the clustering analysis that generates the cluster centroids that VoltView is using for the onsite classification. Moreover, multiple video comparison mode has interactive settings onsite to switch between parameters to define the color coding of increase or decrease of the

chosen parameter in each imaged neuron. The onsite and detailed analysis was written in MATLAB (MathWorks) using custom-written scripts. In principal steps, the recorded videos from '.CXD' files were imported to MATLAB using bioformats v.6.11 package (OME; https://www.openmicroscopy.org/). Most importantly, VoltView is not only filtering out traces that we can visually further explore, but detects APs, EPSPs, IPSPs, Bursts and extracts the 29 different o-phys parameters for classification of PRTs. All the calculated values and features were stored in struct variables so that VoltView can call them for the ROI explorer plotting and for the onsite analysis. For more details, see the user's guide at https://zenodo.org/record/8030176.

**Classification and clustering of o-phys.** Parameters for hierarchical clustering were extracted from the o-phys traces by custom-written routines in MATLAB (MathWorks). Parameters were averages of six to seven sweeps recorded from each ROI. 'o-EPSP' and 'o-IPSP' was the number of o-PSPs during the 20 Hz Op. 'Paired Pulse 2-1' compared the second o-PSP with the first o-PSP amplitude. 'Paired Pulse 3-2' compared the third o-PSP with the second o-PSP amplitude. 'Subthr Slope q2-q1' compared the mean amplitude of o-Sub between the second 250 ms and the first 250 ms. 'Subthr Slope q4-q1' compared the mean amplitude of o-Sub between the last 250 ms and the first 250 ms. 'AP onset' is the delay of the first o-AP during 20 Hz Op. 'AP bimodality coeff' calculated Sarle's bimodality coefficient as the square of skewness divided by the kurtosis; the value for uniform distribution is 5/9, values greater than that indicate bimodal (or multimodal) distribution. 'AP bimodality binary' gave 1 for bimodal (when coefficient was above 5/9) and 0 for nonbimodal o-AP peak distribution. 'AP % in burst' quantified the number of o-APs inside burst periods. 'Burst AP freq (Op)' quantified the firing rate inside detected bursts during 20 Hz Op. 'AP freq 2nd/1st (Op)' quantified the change of firing rate during the 20 Hz Op comparing the second half (Op-2) with the first half (Op-1). 'Burst AP freq (Post)' quantified the firing rate inside detected bursts after 20 Hz Op. 'AP freq 2nd/1st (Post)' quantified the change of firing rate after the 20 Hz Op comparing the second half (Post-2) with the first half (Post-1). 'AP num' gave the total number of o-APs detected on the baseline of all six to seven sweeps. 'AP freq (Baseline)' was the average firing rate on the baseline before the 20 Hz Op. 'AP freq (Op-1)' was the average firing rate on the first half of 20 Hz Op. 'AP freq (Op-2)' was the average firing rate on the second half of 20 Hz Op. 'AP freq (Post-1)' was the average firing rate on the first half of Post, after 20 Hz Op. 'AP freq (Post-2)' was the average firing rate on the second half of Post, after the 20 Hz Op. 'Burst num (Op-1)' was the number of burst periods during the first half of the 20 Hz Op. 'Burst num (Op-2)' was the number of burst periods during the second half of the 20 Hz Op. 'Burst num (Post-1)' was the number of burst periods during the first half of Post, after the 20 Hz Op. 'Burst num (Post-2)' was the number of burst periods during the second half of Post, after the 20 Hz Op. 'Burst length (Op-1)' was the average length of burst periods during the first half of 20 Hz Op. 'Burst length (Op-2)' was the average length of burst periods during the second half of 20 Hz Op. 'Burst length (Op-2/Op-1)' was the change of burst length from Op-1 to Op-2. 'Burst length (Post-1)' was the average length of burst periods during the first half of Post, after the 20 Hz Op. 'Burst length (Post-2)' was the average length of burst periods during the second half of Post, after the 20 Hz Op. 'Burst length (Post-2/Post-1)' was the change in burst length from Post-1 to Post-2. Agglomerative clustering was done by Ward's algorithm, Euclidean distance for the cutoff and for definition of the number of clusters was chosen based on the slope-change of number of clusters versus the Euclidean distance cutoff.

**VoltView classifier.** We used the unbiased hierarchical clustering of the connectome dataset and calculated the cluster centroids for each o-phys cluster. We had all 29 parameters in the cluster centroid. We used two parallel classifications and used their consensus. First, for each neuron we calculated the square root of sum squared differences

# Article

compared with all the cluster centroids and ranked the distances to choose the closest three clusters for each neuron. Next, for each neuron we calculated the level of correlation to all the cluster centroids, ranked the correlation and choose the three highest correlating clusters. If the consensus of the two parallel classifications agreed on one or more clusters, we compared the ranks and assigned the closest cluster to each neuron.

**Spatial cluster density.** The observed minimal distances in space from a reference cell to a cell in the target cluster were calculated, as well as the expected minimal distances by chance, based on 1,000 shuffles of cluster identity, for all cells. The distance metric was calculated for each cell by:

$$\frac{\text{Observed} - \frac{1}{n}\sum_{n=1}^{1000}\text{expected}(n)}{\text{Observed} + \frac{1}{n}\sum_{n=1}^{1000}\text{expected}(n)}$$

and is in range −1:1, where negative values indicated that the minimal distance of the reference cell to a cell in the target cluster is smaller than by chance and positive values indicated greater distance than by chance.

### Reporting summary

Further information on research design is available in the Nature Portfolio Reporting Summary linked to this article.

### Data availability

Voltage-Seq single-cell RNA-sequencing data is available at https://www.ebi.ac.uk/biostudies/arrayexpress with accession code E-MTAB-13104. Example all-optical VMH-PAG Voltron recordings are included with the MATLAB code of VoltView v.1.0 analysis at https://zenodo.org/record/8030176. Source data are provided with this paper.

### Code availability

VoltView v.1.0 used to analyze all-optical voltage imaging and to select neurons for Voltage-Seq neuronal soma harvesting is available at https://zenodo.org/record/8030176 under Creative Commons Attribution v.4.0 International Public License (with installation guide, user's guide and example data).

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

### Acknowledgements

We thank D. Calvigioni, J. Szabadics, M. Rózsa, G. Oláh and H. Brünner, for many helpful discussions and for their technical expertise, K. Meletis for support and A. Wolthon for help with technical service in mouse work. J.F. acknowledges funding from the Swedish Research Council (Starting Grant in Medicine no. 2019-02052), StratNeuro (Start-up program 2020), Hjärnfonden (grant nos. FO2020-0162 and FO2022-0323), BBRF NARSAD Young Investigator Grant (no. 29079), Jeanssons Foundation (grant no. J2020-0122) and Åke Wiberg Foundation (grant nos. M21-0220 and M22-0223). B.R. acknowledges funding from the Swedish Society for Medical Research (grant no. CG-22-0260-H-02), the Swedish Research Council (grant no. 2022-01620) and the Knut & Allice Wallenberg Foundation (grant no. 2021.0142).

### Author contributions

J.F. and V.C. designed and carried out voltage imaging and patch-clamp electrophysiology experiments. M.H.B. performed histology. J.F. wrote the video analysis. J.F. and V.C. performed the analysis of all-optical voltage imaging and patch-clamp electrophysiology data. J.F. and V.C. performed surgeries and viral injections. B.R. and J.N. performed single-cell RNA-sequencing and processed transcriptome data. J.F., V.C. and M.H.B. prepared the figures. J.F. wrote the paper.

### Funding

### Competing interests

The authors declare no competing interests.

### Additional information

**Extended data** is available for this paper at https://doi.org/10.1038/s41592-023-01965-1.

**Correspondence and requests for materials** should be addressed to János Fuzik.

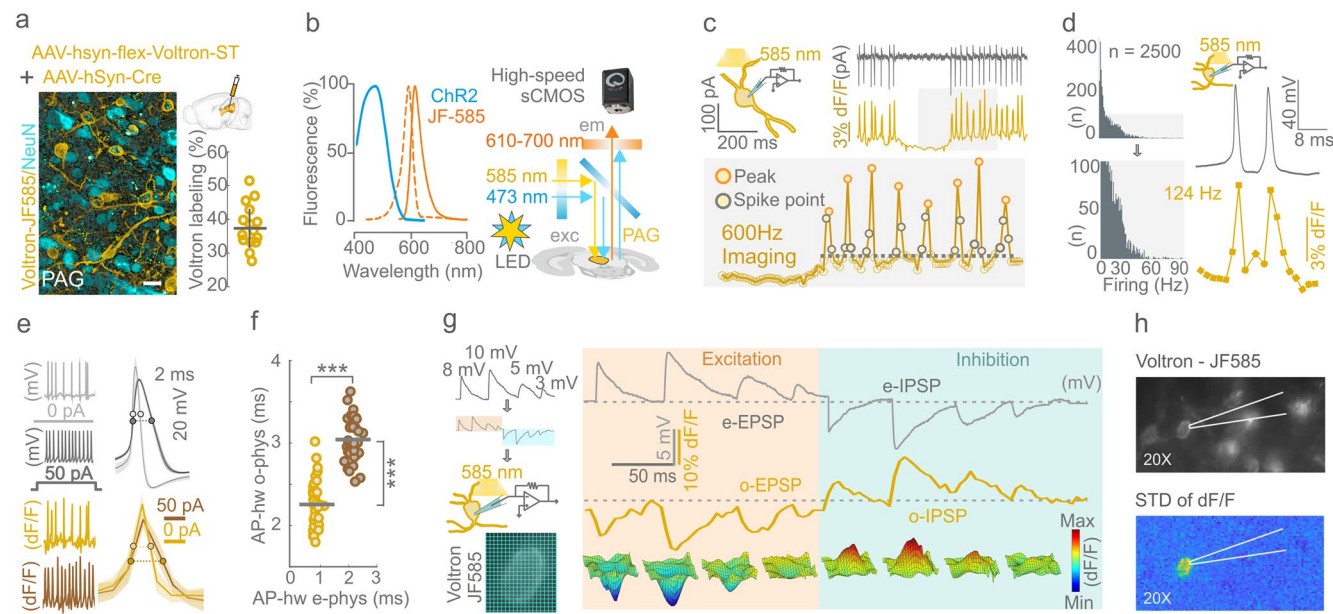

**Extended Data Fig. 1 | Optical setup, frame rate and detection limits of Voltron imaging.** All Voltron traces were reversed except in panel g, 473 nm was 2.5 mW/mm2 and 585 nm power was 14 mW/mm2. **a**, NeuN staining and Voltron labeling with JF-585 (left, Scale bar is 20 μm). Scheme of Voltron-expressing AAV injection to PAG, hive plot (mean ± sd) of the percentage of JF-585-Voltron-ST neurons/total number of neurons in randomly designated areas of PAG (17 areas, 6 slices, N = 2). **b**, Excitation, and emission profile of ChR2 (blue) and JF-585 (orange) (left). Optical setup (right). **c**, Scheme of simultaneous cell-attached patch-clamp recording and voltage imaging (top left), APs from a PAG neuron with time-aligned traces of e-phys (black) and o-phys (gold) from the same neuron (top right). Insert of o-phys trace acquired at 600 Hz framerate with AP peaks (Peak points) and points between AP-threshold and peak (Spike points), dashed line marks AP-threshold (bottom). **d**, Histogram of detected firing rate in 2500 Pag neurons with example trace of the highest firing rate (-124 Hz) detected: (right). **e**, E-phys trace at 0 pA (light gray) and 50 pA (dark gray) depolarizing current injection with the corresponding o-phys traces at 0 pA (gold) and 50 pA (brown) (left). AP shape of e-phys (light gray, dark gray) and o-phys spikes (gold, brown) at 0 pA and 50 pA current injection (right). **f**, Half-width of APs compared (DABEST permutation test, two-sided, ***p < 0.001) across 0 pA and 50 pA current injection of e-phys (DABEST p = 1,9e-49) and o-phys (DABEST p = 6,4e-16) recordings, horizontal bar:mean (N = 2). g, Scheme of symmetric voltage command (8, 10, 5, 3 mV and −8, −10, −5, −3 mV) to test subthreshold sensitivity of the Voltron (top left). Scheme of experiment (middle left). Soma of a JF-585-Voltron-ST neuron, representing the pixel-binning for the surface plots (bottom left) (scale bar, 10 μm). Application of the symmetric voltage command in whole-cell voltage-clamp during voltage imaging, e-phys recording (top, black), with the corresponding o-phys (bottom, gold) trace from the same recorded neuron (avg. of 10 sweeps) (n = 6, N = 3). Surface plots of dF/F of the recorded soma with 4 by 4 binned pixels (bottom). h, JF-585-Voltron-ST neurons during imaging and whole-cell patching in 20× magnification (top). Std of dF/F of the whole-cell-recorded neuron, with the surrounding neurons not depolarized (bottom) (scale bar, 50 μm).

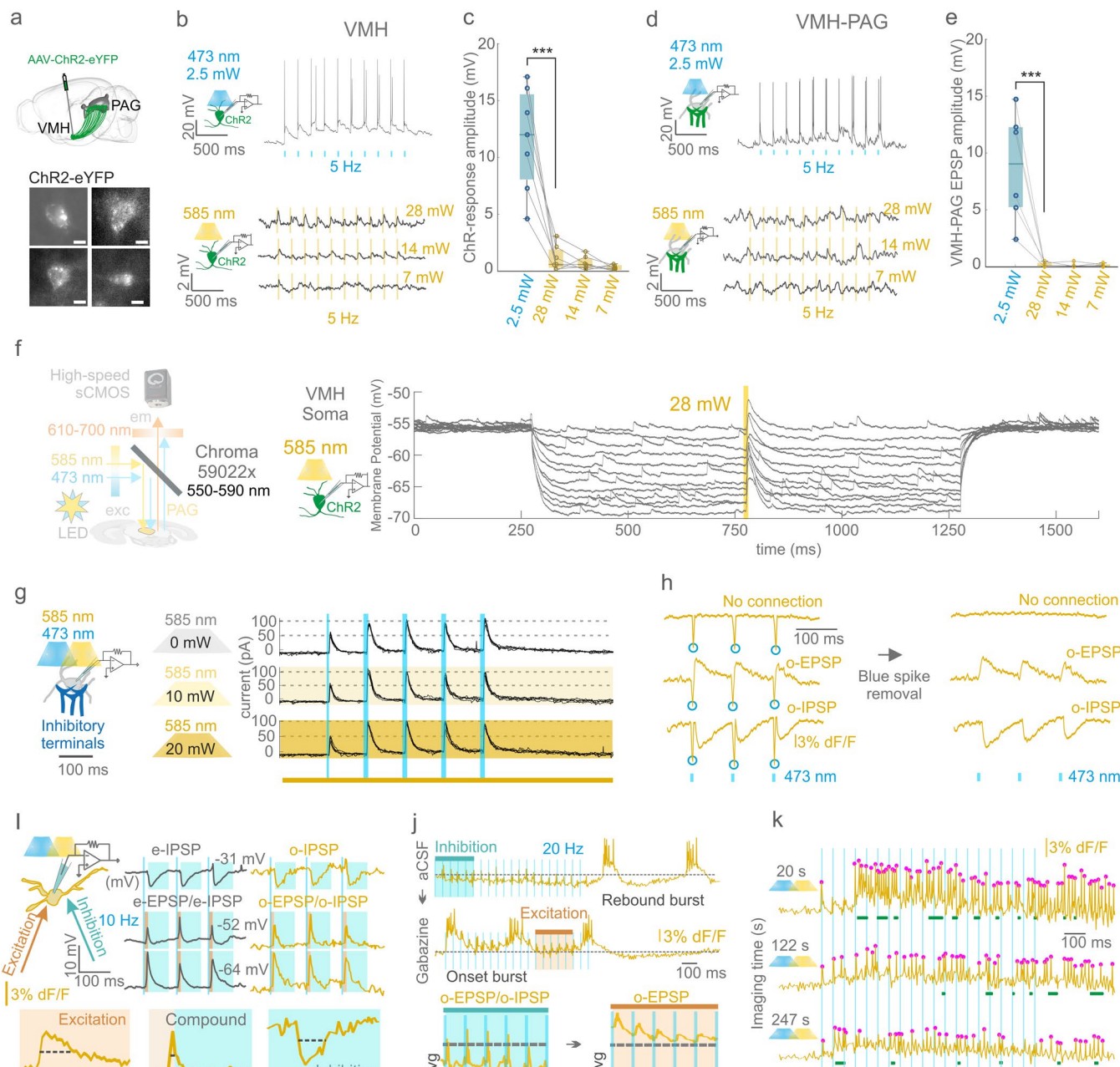

**Extended Data Fig. 2 | Testing Voltron imaging and validation of all-optical recordings.** All Voltron traces were reversed, 473 nm was 2.5 mW/mm2 and 585 nm power was 14 mW/mm2. **a**, Scheme of ChR2 expression in VMH-PAG and ChR2-expressing VMH somas (scale bar 10 μm) (N = 8). **b**, Scheme of the experiment (top left), activation of ChR2 on VMH neuronal soma with 473 nm blue light with 5 Hz at 2.5 mW/mm2 and firing response upon each ChR2 activation (top right); activation of ChR2 on VMH neuronal soma with 585 nm yellow light (bottom left) with 5 Hz at 7, 14, 28 mW/mm2 (bottom right). **c**, Box plot (center=median, box bottom=25th percentile, box top=75th percentile, whiskers do not consider outliers: larger than 75th percentile + interquartile range, or smaller than 25th percentile - interquartile range) of somatic ChR2-response amplitudes upon 473 nm and 585 nm activation, where in firing neurons the amplitude was measured between the holding potential and the AP threshold (473 nm vs 585 nm at 28 mW: t-test, two-sided, ***p < 0.001, p = 0.00063 N = 4, n = 7). **d**, Scheme of experiment with whole-cell-recorded

PAG neuron and optogenetic activation of VMH-PAG terminals with 473 nm blue light with 5 Hz at 2.5 mW/mm2 (top left). Postsynaptic firing response upon each ChR2 activation (top right). attempt to the activation of ChR2 on VMH neuronal soma with 585 nm yellow light (bottom left) with 5 Hz at 7, 14, 28 mW/mm2 (bottom right). **e**, Box plot (center=median, box bottom=25th percentile, box top=75th percentile, whiskers: larger than 75th percentile + interquartile range, or smaller than 25th percentile - interquartile range) of VMH-PAG terminal-evoked postsynaptic response amplitudes upon 473 nm and 585 nm activation, where in firing neurons the amplitude was measured between the holding potential and the AP threshold (473 nm vs 585 nm at 28 mW: t-test, two-sided, ***p < 0.001, p = 0.00000019 N = 4, n = 6). **f**, Scheme of the imaging setup semi-transparent with the 550–590 nm excitation filter in non-transparent black (left) and extreme example of ChR2 activation by 585 nm light pulse at 28 mW/mm2 power that induced ~5 mV depolarizations.

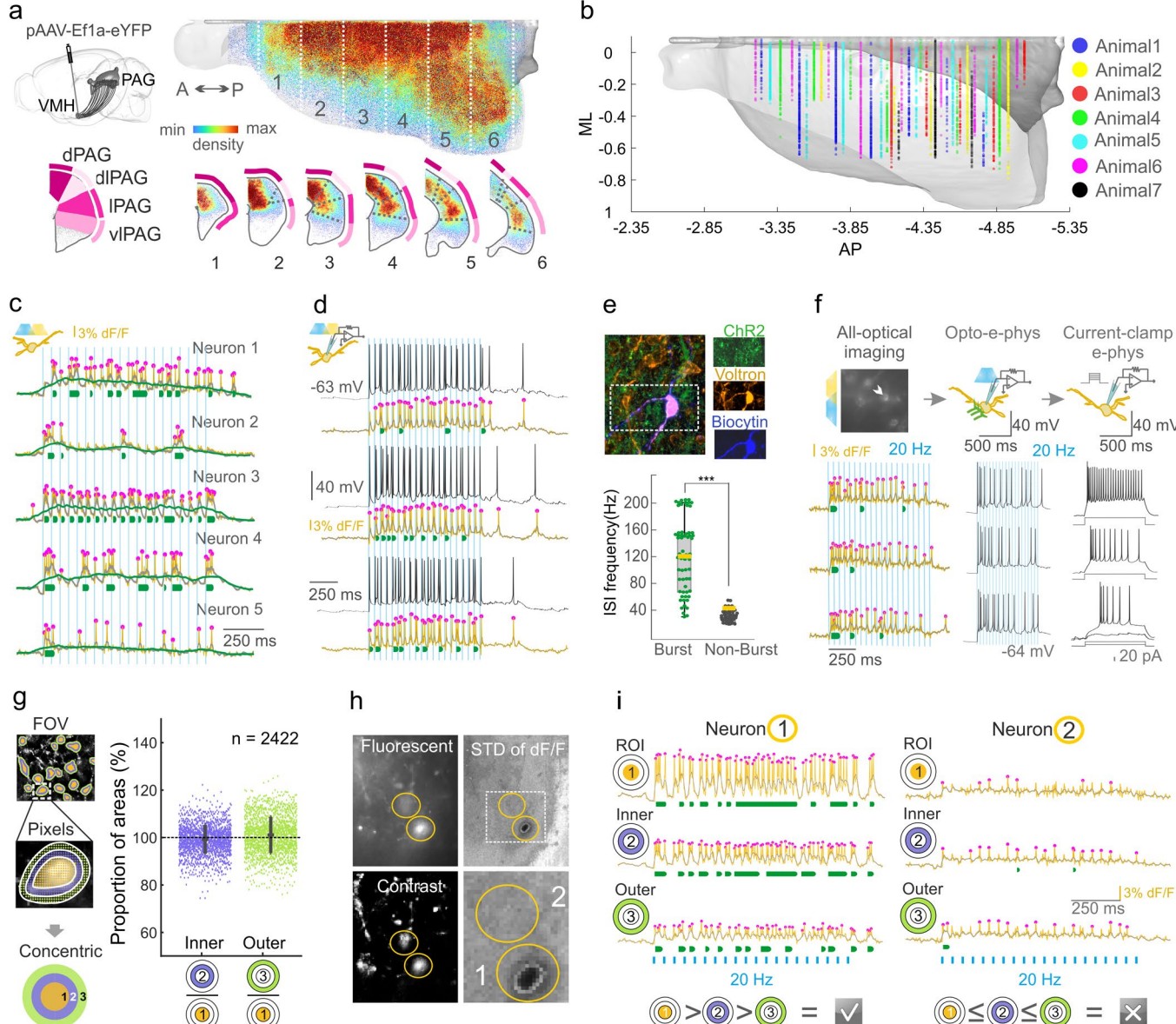

**Extended Data Fig. 3 | Axonal map, Imaged coordinates, Burst detection, Concentric.** All Voltron traces were reversed, 473 nm was 2.5 mW/mm2 and 585 nm power was 14 mW/mm2. **a**, Expression of eYFP in the VMH-PAG pathway (top left); PAG hemisphere with 3D reconstruction of VMH terminal density, color codes density from blue (low) to red (high) (top). Coronal scheme shows the color code of PAG subregions (bottom left); Coronal bins (1: −3,25/−3,65 2: −3,65/−3,95; 3: −3,95/−4,25; 4: −4,25/−4,55, 5: −4,55/−4,85, 6: −4,85/−5,15 mm to Bregma) show distribution of VMH axons across PAG subregions (bottom). **b**, A-P coordinates of imaging planes with 6911 imaged neurons (Dots in the planes are individual neurons (6911 neurons/7 animals). **c**, All-optically imaged neurons (Neuron 1-Neuron5) with different burst types. O-APs (magenta), o-phys traces (gold), o-sub (gray), moving average of 100 points (green). Green lines under o-phys traces indicate bursts (N = 2). **d**, PAG neuron during all-optical imaging and whole-cell recording. Corresponding o-phys (gold) and e-phys (black) sweeps, magenta dots are o-APs, green lines under the o-phys traces indicate bursts. **e**, Biocytin-filled JF-585-Voltron-ST PAG neuron between the ChR2-expressing VMH axonal terminals (top)(scale bar, 50 µm). Box plot (center=median, box bottom=25th percentile, box top=75th percentile, whiskers 75th percentile + interquartile range,

25th percentile - interquartile range) compares the frequency of ISIs where burst was detected (green dots) compared to ISIs where bursts were not detected (gray dots). Dots represent individual ISIs from 6 o-phys sweeps of the shown biocytin labeled neuron (mean ± sd t-test, two-sided, ***p < 0.001, p = 0.00005). **f**, FOV of JF-585-Voltron-ST neurons (top), white arrow: chosen neuron for whole-cell recording that onset-bursted in o-phys (left). Whole-cell recording during 20 Hz Op confirmed the onset-bursting (middle). Current-clamp confirmed a burst firing type by intrinsic features (right). (scale bar, 50 µm) **g**, Concentric made two concentric areas (2: purple (Inner), 3: green (Outer)) around somatic ROIs (1: gold) (left). Scatter plot of surface ratio area (2)/(1) ± 5.2% sd area (3)/(2) ± 8.3% sd) areas of n = 2422 neurons. (scale bar, 20 µm, insert 3 µm) **h**, Fluorescent image of two JF-585-Voltron-ST neurons (yellow circles) (top left), local equalized and contrast enhanced JF-585 signals (yellow circles) (bottom left), ±sd of dF/F voltage signal of the same neurons (yellow circles) (scale bar, 20 µm).**i**, Concentric analysis of the two neurons from h. Neuron 1 displayed strong firing and bursting activity (ROI), which was detected in Inner, Outer with decreased o-AP amplitude (left).

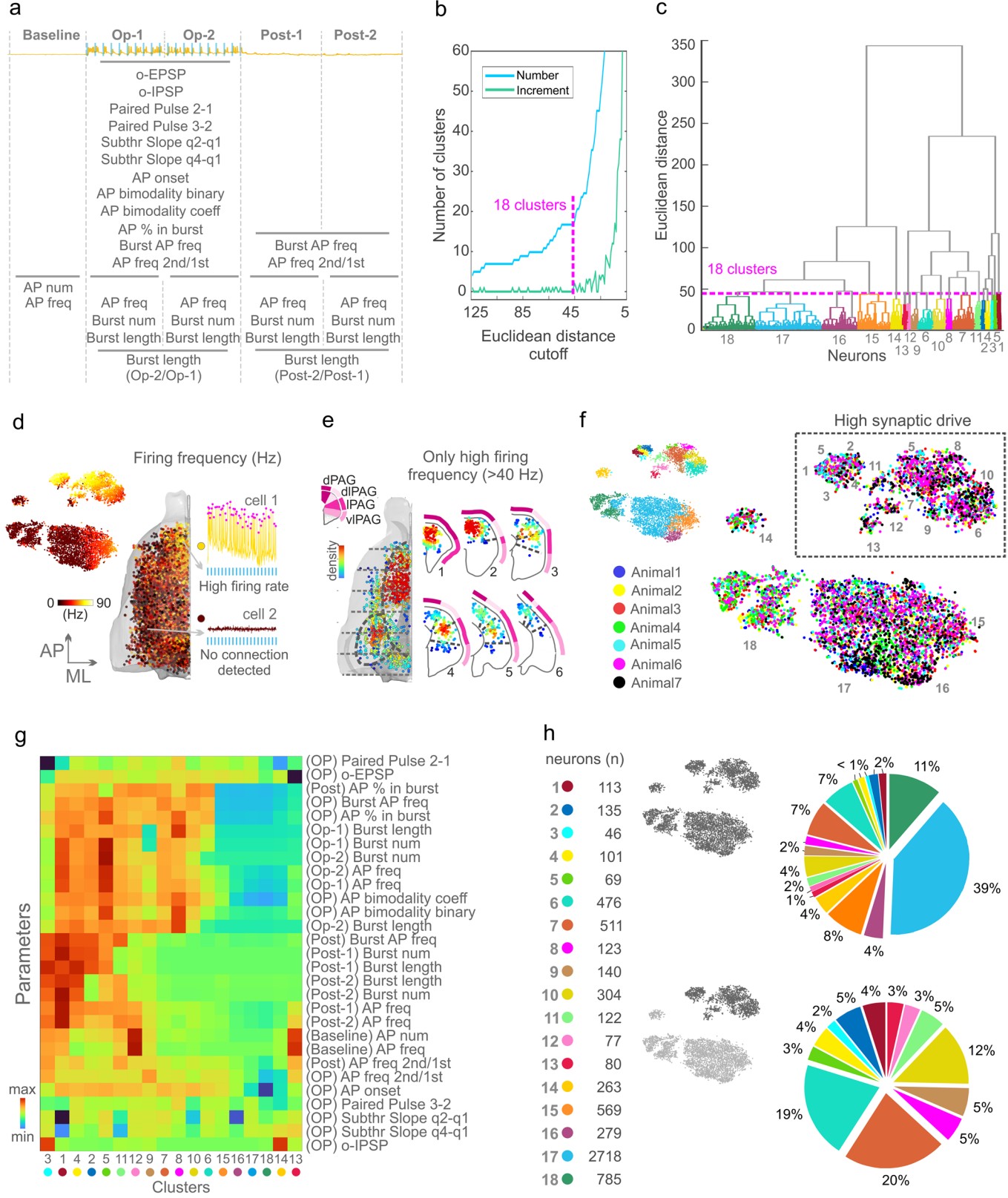

**Extended Data Fig. 4 | See next page for caption.**

**Extended Data Fig. 4 | Clustering of VMH-PAG o-phys PRTs. a**, 29 parameters extracted from the o-phys traces used for the hierarchical clustering of o-phys types. Scheme on top details the temporal segments of all-optical sweeps with Op-1: first half of Op, Op-2: second half of Op, Post-1: first half of after-Op, Post-2: second half of after-Op. **b**, line plot of the relation of number of clusters versus Euclidean distance cutoff of the agglomerative hierarchical clustering (blue) and the first derivative of the same curve showing the increase in the number of clusters versus Euclidean distance cutoff of the agglomerative hierarchical clustering (green). **c**, Hierarchical dendrogram of o-phys of the 6911 neurons with the Euclidean cutoff resulting in 18 clusters (color code matches Fig. 2f, g). **d**, t-SNE plot of VMH-PAG o-phys clusters with color coding the imaged neurons by the average firing frequency during Op (left), 3D map spatial distribution of the imaged neurons color coded by the average firing frequency during Op (right) with 'cell1' example of high firing (84 Hz) and 'cell2' where no connection was detected. **e**, Spatial mapping of neurons in the VMH-PAG connectome which responded with high firing frequency ( < 40 Hz) (left), and the anatomical mapping of the same neurons in the coronally sliced PAG hemisphere (right) coronal bins (1: −3,25/−3,65 2: −3,65/−3,95; 3: −3,95/−4,25; 4: −4,25/−4,55, 5: −4,55/−4,85, 6: −4,85/−5,15 mm to Bregma) **f**, All-optical voltage-imaged VMH-PAG with 'High synaptic drive' clusters highlighted in rectangle, to show the lack of batch effect across t-SNE regions and clusters (n = 6911, N = 7). **g**, Heat map of clustering parameters versus the 18 o-phys clusters show characteristics of the distinct o-phys clusters (N = 7). **h**, Quantification of neurons in each o-phys cluster (left). Pie chart of cluster proportions (%) in all the VMH-PAG connectome (top right), and in the high synaptic drive clusters (bottom right) (N = 7).

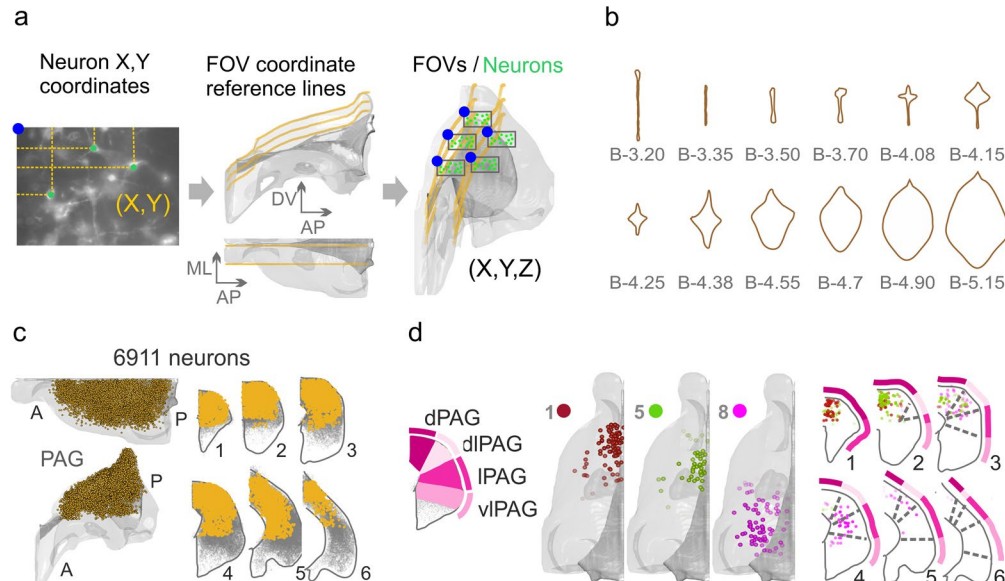

**Extended Data Fig. 5 | Spatial mapping of all-optical connectome. a**, 3D mapping of the all-optical voltage-imaged PAG neurons; FOV with JF-585-Voltron-ST neurons illustrate the extraction of X and Y coordinates relative to the top left corner of each FOV (blue) (left), reference lines were spaced away by the size of the FOVs (middle) and was used to rail the FOVs upon mapping up to the identified A-P coordinate (right)(Scale bar, 50 μm). **b**, Line drawings of the aqueduct (AQ) along the A-P axis of PAG, templated with coronal PAG sections of the Allen reference atlas. A-P coordinate was defined based on the shape

of the AQ **c**, 3D hemisphere of PAG with 6911 neurons (gold) dorsal view (top), transverse view (bottom) (left), Coronal bins (1: −3,25/−3,65 2: −3,65/−3,95; 3: −3,95/−4,25; 4: −4,25/−4,55, 5: −4,55/−4,85, 6: −4,85/−5,15 mm to Bregma) show the coverage of PAG subregions (right). **d**, 3D PAG model of one hemisphere for v-phys cluster 1,5,8 with the density-core mapping of postsynaptic neurons colored by cluster identity (left), merging the spatial coordinates of cluster 1,5 and 8 on the same coronal bins as in c (right).

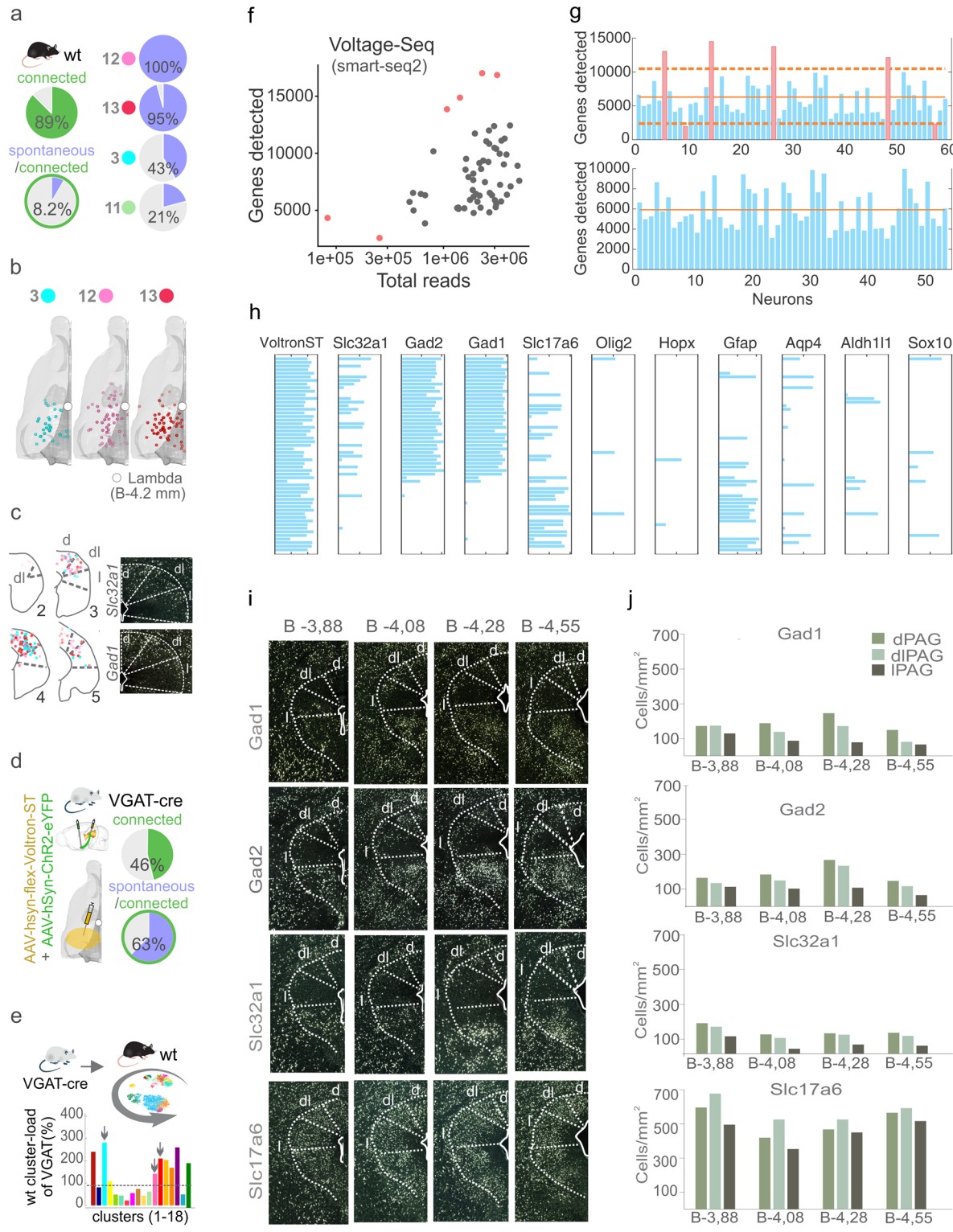

**Extended Data Fig. 6 | See next page for caption.**

**Extended Data Fig. 6 | Spontaneously active clusters, scRNA-seq, ISH quantification. a**, Pie chart of connections in wt mouse (top left), and pie chart of spontaneously firing connected neurons (bottom left). Pie charts of the ratio of spontaneously active neurons within cluster 12, 13, 3, 11 (right). **b**, 3D PAG with the density-core mapping of neurons of o-phys cluster 3,12,13; (white circle indicates Bregma-4.2 mm). **c**, Coronal bins of PAG (2: 3,65/3,95; 3:3,95/4,25; 4: 4,25/4,55, 5: 4,55/4,85) merging the spatial coordinates of cluster 3,12,13 (left), spatial distribution of *Slc32a1* and *Gad1* ISH signal (modified from the Allen ISH database) (right)(Scale bar, 500 μm). **d**, Scheme of ChR2 and Voltron-ST expression in the posterior PAG of VGAT-Cre mice (left) (N = 4), pie chart of detected connections in VGAT-cre connectome (top right), and pie chart of spontaneously firing postsynaptic neurons within the inhibitory VMH-PAG connectome (bottom right). Spontaneous firing was 8-times higher (63.7%) in the VGAT data **e**, VGAT PRTs were classified by VoltView to probe cluster-load of wild-type clusters with VGAT o-phys. Cluster 1,3,12,13 and subthreshold clusters (14,15,16,18) had the highest relative cluster-load with VGAT o-phys. **f**, Scatter plot of detected genes/number of reads with Smart-Seq2 in Voltage-Seq data. **g**, Bar plot of number of detected genes in each 60 Voltage-Seq neurons (blue), mean (solid orange) and ±1.5 × sd (dashed orange), 6 discarded neurons with red bars (top); Bar plot of number of detected genes in quality controlled 54 Voltage-Seq neurons (blue) with new mean (solid orange) (bottom). **h**, Bar plot of Voltage-Seq RNA-transcriptome with the expression of *Voltron-ST, Slc32a1, Gad2, Gad1, Slc17a6 and glial markers Olig2, Hopx, Gfap, Agp4, Aldh1l1, Sox10*. Minor ticks are 5 and 10 TPM from left to right for each gene. **i**, ISH of PAG coronal slices (−3.88, −4.08, −4.28, −4.55 from Bregma) implemented from the Allen ISH database with inverted colors. GABAergic markers: *Gad1* (GAD67), *Gad2* (GAD65), *Slc32a1* (VGAT), and glutamatergic marker: *Slc17a6* (VGLUT2) shows the spatial distribution of GABAergic and glutamatergic neurons, respectively (top to bottom)(Scale bar, 500 μm). **j**, Bar plots of ISH quantification (positive nuclei/mm²) in dPAG, dlPAG and lPAG in four A-P coordinates (−3.88, −4.08, −4.28, −4.55 from Bregma) for *Gad1, Gad2, Slc32a1*, and *Slc17a6* (top to bottom).

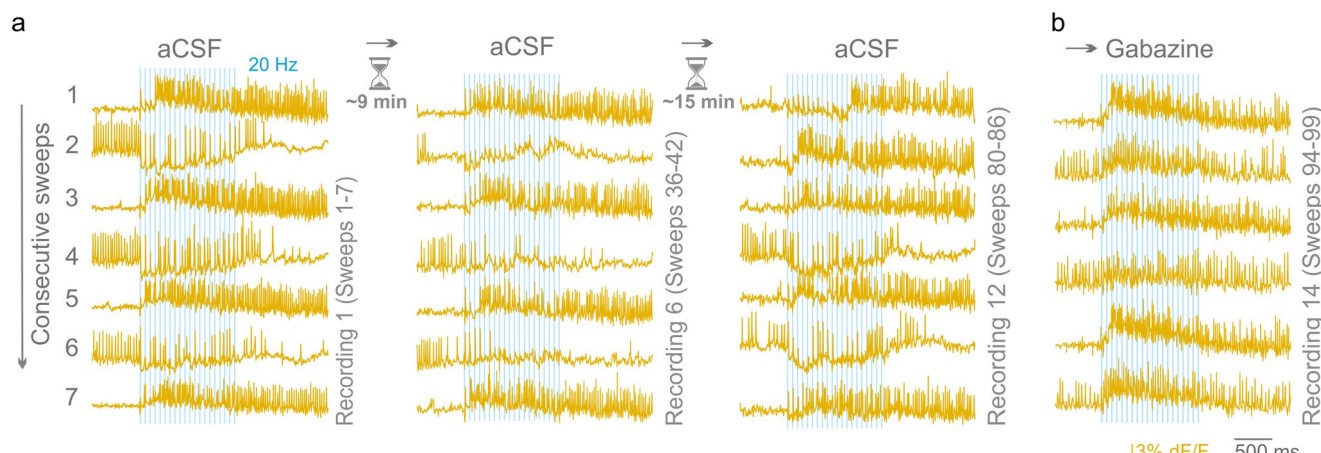

**Extended Data Fig. 7 | Switch response.** All Voltron traces were reversed, 473 nm was 2.5 mW/mm2 and 585 nm power was 14 mW/mm2. **a**, Example o-phys traces of the same Switch responder over time. First all-optical recording (left), all-optical recording 9 min later (middle), and all-optical recording 15 min after the second recording (26 min in the recording chamber). Switch behavior could be observed throughout more than 80 sweeps with a stable switching using our recording conditions. **b**, consecutive sweeps of a Switch response in Gabazine where all sweeps turned excitatory.

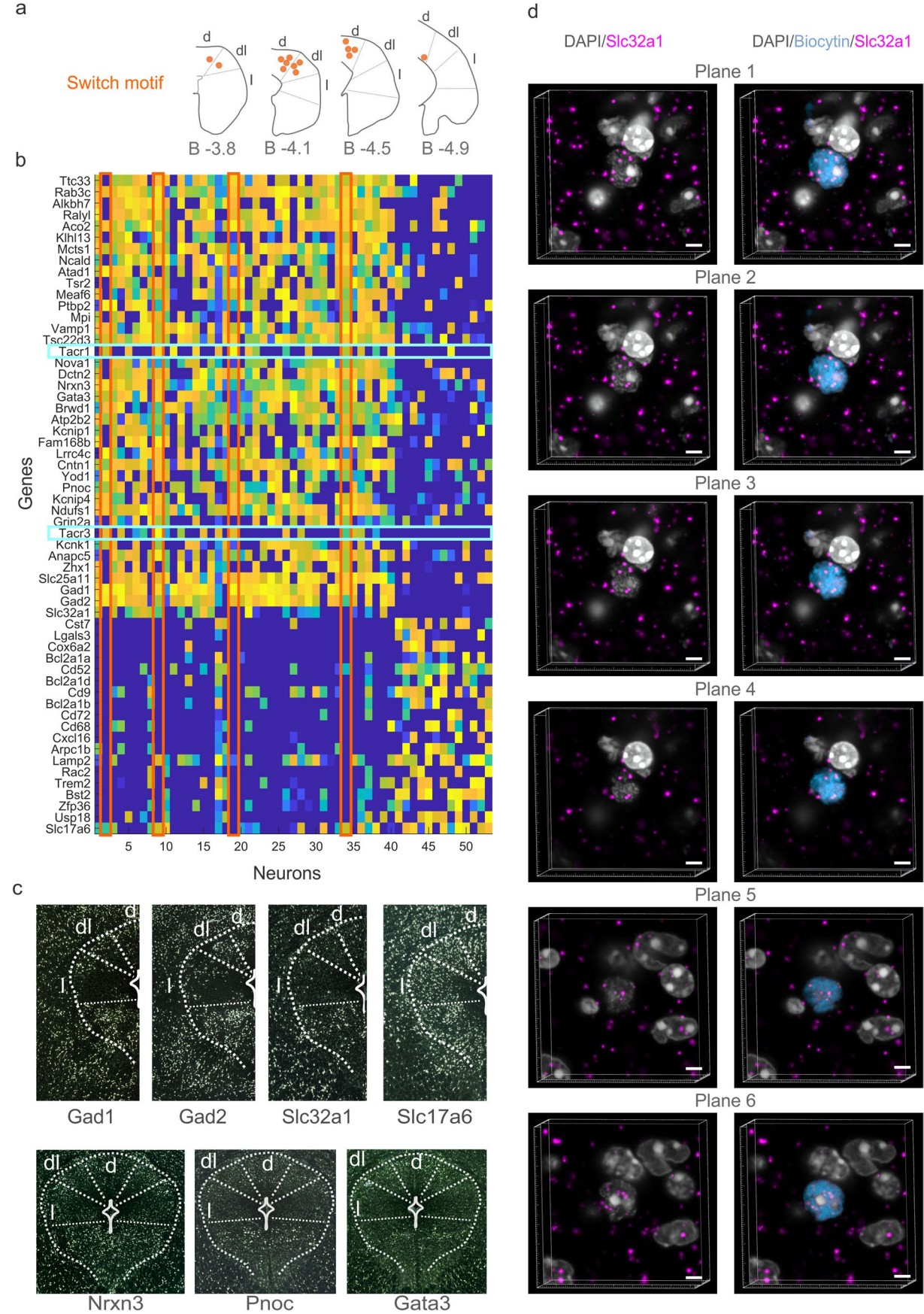

**Extended Data Fig. 8 | See next page for caption.**

**Extended Data Fig. 8 | Differential expression, GABAergic markers, ISH.**
**a**, Anatomical position of Switch responders (orange dots) on coronal slices in d-, dl-, lPAG. **b**, Heatmap of DE genes; Switch responders (orange rectangles), express *Tacr1* and *Tacr3* (cyan rectangles). **c**, ISH of PAG coronal slices at B-4.08 implemented from the Allen ISH database with inverted colors. *Gad1*, *Gad2*, *Slc32a1*, and *Slc17a6* shows the spatial distribution of GABAergic and glutamatergic neurons, respectively (top); ISH of PAG coronal slices with DE-identified putative GABAergic markers (*Nrxn3, Pnoc, Gata3*) shows spatial distribution of putative marker expression (Scale bar, 500 μm). **d**, 6 different planes of *Slc32a1 (magenta)* ISH on an example biocytin-filled burst firing Switch responder shows nuclear colocalization of *Slc32a1* (scale bar, 2 μm) (N = 4).

# Reporting Summary

## Statistics

For all statistical analyses, confirm that the following items are present in the figure legend, table legend, main text, or Methods section.

| n/a | Confirmed | |
|---|---|---|
| ☐ | ☒ | The exact sample size (*n*) for each experimental group/condition, given as a discrete number and unit of measurement |
| ☐ | ☒ | A statement on whether measurements were taken from distinct samples or whether the same sample was measured repeatedly |
| ☐ | ☒ | The statistical test(s) used AND whether they are one- or two-sided<br>*Only common tests should be described solely by name; describe more complex techniques in the Methods section.* |
| ☒ | ☐ | A description of all covariates tested |
| ☒ | ☐ | A description of any assumptions or corrections, such as tests of normality and adjustment for multiple comparisons |
| ☐ | ☒ | A full description of the statistical parameters including central tendency (e.g. means) or other basic estimates (e.g. regression coefficient) AND variation (e.g. standard deviation) or associated estimates of uncertainty (e.g. confidence intervals) |
| ☐ | ☒ | For null hypothesis testing, the test statistic (e.g. *F*, *t*, *r*) with confidence intervals, effect sizes, degrees of freedom and *P* value noted<br>*Give P values as exact values whenever suitable.* |
| ☒ | ☐ | For Bayesian analysis, information on the choice of priors and Markov chain Monte Carlo settings |
| ☐ | ☒ | For hierarchical and complex designs, identification of the appropriate level for tests and full reporting of outcomes |
| ☒ | ☐ | Estimates of effect sizes (e.g. Cohen's *d*, Pearson's *r*), indicating how they were calculated |

*Our web collection on statistics for biologists contains articles on many of the points above.*

## Software and code

Policy information about availability of computer code

| Data collection | HCImage Live 4.6.0, pClamp 10.4, Matlab 2021b |
|---|---|
| Data analysis | Matlab 2021b; VoltView 1.0 (https://zenodo.org/record/8030176) |

For manuscripts utilizing custom algorithms or software that are central to the research but not yet described in published literature, software must be made available to editors and reviewers. We strongly encourage code deposition in a community repository (e.g. GitHub). See the Nature Portfolio guidelines for submitting code & software for further information.

## Data

Policy information about availability of data

All manuscripts must include a data availability statement. This statement should provide the following information, where applicable:

- Accession codes, unique identifiers, or web links for publicly available datasets
- A description of any restrictions on data availability
- For clinical datasets or third party data, please ensure that the statement adheres to our policy

We gave access to "minimum dataset" to test our VoltView 1.0 analysis, we made example raw all-optical voltage imaging data available at https://zenodo.org/record/8030176 , and we included source data for Figure 2 and Figure 3 with the manuscript. We deposit RNA-transcriptomic data of Voltage-Seq experiments to https://www.ebi.ac.uk/fg/annotare/help/submit_exp.html

## Human research participants

Policy information about studies involving human research participants and Sex and Gender in Research.

| | |
|---|---|
| Reporting on sex and gender | N/A |
| Population characteristics | N/A |
| Recruitment | N/A |
| Ethics oversight | N/A |

Note that full information on the approval of the study protocol must also be provided in the manuscript.

# Field-specific reporting

Please select the one below that is the best fit for your research. If you are not sure, read the appropriate sections before making your selection.

☒ Life sciences  ☐ Behavioural & social sciences  ☐ Ecological, evolutionary & environmental sciences

For a reference copy of the document with all sections, see nature.com/documents/nr-reporting-summary-flat.pdf

# Life sciences study design

All studies must disclose on these points even when the disclosure is negative.

| | |
|---|---|
| Sample size | For our largest data body we used 7 mice, and voltage imaged ~7000 neurons in them. We continued the experiments until we had sufficient spatial coverage of the voltage-imaged area (at least 40 cells/200umx200umx200um tissue volumes). |
| Data exclusions | No data were excluded. |
| Replication | Multiple animals were voltage-imaged at the same XYZ coordinates in the brain and results were cross-compared to validate the lack of batch-effect. Experiments were done in the period of ~4 months, and were replicated every 1-2 weeks successfully depending on sufficient expression of the voltage sensor in the neurons to image. |
| Randomization | We conducted mainly proof of principal experiments without multiple groups, thus comparisons and randomization were not crucial |
| Blinding | Blinding was not relevant during data collection because we conducted experiments in the same manner, tile-imaged the same coordinates to cover the same brain areas. During analysis, the same analysis script was used to all the data, blinding was not relevant either. |

# Reporting for specific materials, systems and methods

We require information from authors about some types of materials, experimental systems and methods used in many studies. Here, indicate whether each material, system or method listed is relevant to your study. If you are not sure if a list item applies to your research, read the appropriate section before selecting a response.

### Materials & experimental systems

| n/a | Involved in the study |
|---|---|
| ☐ | ☒ Antibodies |
| ☒ | ☐ Eukaryotic cell lines |
| ☒ | ☐ Palaeontology and archaeology |
| ☐ | ☒ Animals and other organisms |
| ☒ | ☐ Clinical data |
| ☒ | ☐ Dual use research of concern |

### Methods

| n/a | Involved in the study |
|---|---|
| ☒ | ☐ ChIP-seq |
| ☒ | ☐ Flow cytometry |
| ☒ | ☐ MRI-based neuroimaging |

## Antibodies

| | |
|---|---|
| Antibodies used | NeuN (Millipore, MAB377, clone A60, 1:1000 dil); Alexa Fluor 633-conjugated streptavidin (Thermo Fisher, S21375, 1:1000 dil); Cy™5 AffiniPure Donkey Anti-Mouse IgG (H+L) (Jackson, Code:715-175-151, 1:500 dil) |
| Validation | Anti-NeuN Antibody, clone A60 detects level of NeuN and has been published and validated for use in FC, IC, IF, IH, IH(P), IP and WB |

| Validation | (https://www.merckmillipore.com/SE/en/product/Anti-NeuN-Antibody-clone-A60,MM_NF-MAB377). |

# Animals and other research organisms

Policy information about studies involving animals; ARRIVE guidelines recommended for reporting animal research, and Sex and Gender in Research

| Laboratory animals | All mice were 3-5 month old; wild-type: C57BL/6J (Charles River Laboratories), VGAT-Cre: B6J.129S6(FVB)-Slc32a1tm2(cre)Lowl/MwarJ, Jackson stock no. 028862. |
| Wild animals | No wild animals were used to produce the data of the manuscript. |
| Reporting on sex | We used males to minimize potential data variance rising from the sex-differences, because we used fewer animals than for a behavioral or physiological study to further emphasize the power of throughput of our methodology. |
| Field-collected samples | No field-collected samples were used in the study. |
| Ethics oversight | All procedures were approved and performed in accordance and compliance with the guidelines of the Stockholm Municipal Committee (approval no. N166/15 and 7362-2019). |

Note that full information on the approval of the study protocol must also be provided in the manuscript.

