## [Peer Review File · Nature Methods]

Peer Review Information

Manuscript Title: Voltage-Seq: all-optical postsynaptic connectome-guided single-cell transcriptomics

Corresponding author name(s): Janos Fuzik

Editorial Notes:

Redactions – unpublished data	Parts of this Peer Review File have been redacted as indicated to maintain the confidentiality of unpublished data.
Redactions – published data	Parts of this Peer Review File have been redacted as indicated to remove third-party material.

Reviewer Comments & Decisions:

Decision Letter, initial version:

Dear Dr Fuzik,

Thank you for your patience. Your Article, "Voltage-Seq: all-optical postsynaptic connectome-guided single-cell transcriptomics", has now been seen by two reviewers. As you will see from their comments below, although the reviewers find your work of considerable potential interest, they have raised a number of concerns. We are interested in the possibility of publishing your paper in Nature Methods, but would like to consider your response to these concerns before we reach a final decision on publication. We therefore invite you to revise your manuscript to address these concerns.

Please note that we asked for input from an expert in patch-seq, but this reviewer has unfortunately not delivered a review. We may solicit further input from such an expert at a later stage.

[Redacted] This URL links to your confidential home page and associated information about manuscripts you may have submitted, or that you are reviewing for us. If you wish to forward this email to co-authors, please delete the link to your homepage.

We hope to receive your revised paper within 6 weeks. If you cannot send it within this time, please let us know. In this event, we will still be happy to reconsider your paper at a later date so long as nothing similar has been accepted for publication at Nature Methods or published elsewhere.

OPEN SCIENCE REQUIREMENTS

REPORTING SUMMARY AND EDITORIAL POLICY CHECKLISTS

Please note that these forms are dynamic ‘smart pdfs’ and must therefore be downloaded and completed in Adobe Reader. We will then flatten them for ease of use by the reviewers. If you would like to reference the guidance text as you complete the template, please access these flattened versions at <http://www.nature.com/authors/policies/availability.html>.

DATA AVAILABILITY

All novel DNA and RNA sequencing data, protein sequences, genetic polymorphisms, linked genotype and phenotype data, gene expression data, macromolecular structures, and proteomics data must be deposited in a publicly accessible database, and accession codes and associated hyperlinks must be provided in the “Data Availability” section.

Please include a “Data availability” subsection in the Online Methods. This section should inform readers about the availability of the data used to support the conclusions of your study, including accession codes to public repositories, references to source data that may be published alongside the paper, unique identifiers such as URLs to data repository entries, or data set DOIs, and any other statement about data availability. At a minimum, you should include the following statement: “The data that support the findings of this study are available from the corresponding author upon request”, describing which data is available upon request and mentioning any restrictions on availability. If DOIs are provided, please include these in the Reference list (authors, title, publisher (repository name), identifier, year). For more guidance on how to write this section please see: <http://www.nature.com/authors/policies/data/data-availability-statements-data-citations.pdf>

CODE AVAILABILITY

Please include a “Code Availability” subsection in the Online Methods which details how your custom code is made available. Only in rare cases (where code is not central to the main conclusions of the paper) is the statement “available upon request” allowed (and reasons should be specified).

For more information on our code sharing policy and requirements, please see: <https://www.nature.com/nature-research/editorial-policies/reporting-standards#availability-of-computer-code>

MATERIALS AVAILABILITY

SUPPLEMENTARY PROTOCOL

To help facilitate reproducibility and uptake of your method, we ask you to prepare a step-by-step Supplementary Protocol for the method described in this paper. We encourage authors to share their step-by-step experimental protocols on a protocol sharing platform of their choice and report the protocol DOI in the reference list. Nature Portfolio 's Protocol Exchange is a free-to-use and open resource for protocols; protocols deposited in Protocol Exchange are citable and can be linked from the published article. More details can found at www.nature.com/protocolexchange/about.

ORCID

Nature Methods is committed to improving transparency in authorship. As part of our efforts in this direction, we are now requesting that all authors identified as 'corresponding author' on published papers create and link their Open Researcher and Contributor Identifier (ORCID) with their account on the Manuscript Tracking System (MTS), prior to acceptance. This applies to primary research papers only. ORCID helps the scientific community achieve unambiguous attribution of all scholarly contributions. You can create and link your ORCID from the home page of the MTS by clicking on 'Modify my Springer Nature account'. For more information please visit please visit www.springernature.com/orcid.

Best regards,
Nina

Nina Vogt, PhD
Senior Editor
Nature Methods

Reviewers' Comments:

Reviewer #1:

Remarks to the Author:

In this work, Csillag et al demonstrate a new methodology to rapidly classify neurons based on their postsynaptic response properties and also to aid transcriptomic interrogation. Authors did commendable effort to achieve this task and provide very impressive datasets to support their claim. They have used soma-targeted Voltron in the connected postsynaptic neurons to achieve all-optical voltage imaging. While this work clearly novel and innovative in combining two different technologies (Voltron and SmartSeq), it is not clear how easily many of the neuroscience labs can utilize the methods that being proposed without spending significant amount of resources and troubleshooting to get this technique established. Though it is clearly step up over patchseq, technical details necessary to reproducibly implement this technique is immensely challenging.

Major concerns:

1. Voltage imaging identified 18 unique clusters in the PAG based on their response characteristics, but how many of them can be reliably captured with patch clamp electrophysiology? What is the ground truth? Do these clusters exist based on their electrophysiological profile?
2. For spatial mapping of the axons in the PAG, were you using slices (250-300um) that were taken for physiology or you imaged thinner slices(50um)?
3. For posthoc extraction of the XYZ coordinates from imaged neurons, were the slices cleared? What microscope were you using to image these neurons for posthoc spatial analysis?
4. How many of the clusters you have seen in Fig. 3 can be also seen with Voltview?
5. Is Voltview a GUI? How user friendly is the software to use? Your current evidence of implanting Voltview for sequencing experiments is 60 neurons from 3 wildtype mice. Not sure how well this works and can be used by extensive neuroscience labs.
6. Cluster 3,13 and 12, were denser in the posterior PAG in our data (Extended Data Fig. 5b) and localized in the d-dIPAG. It's not clear what your definition of posterior PAG mean? Bregma-4.39 onwards?
7. When Voltage-seq was implemented did you discover what is unique about Cluster 3,13 and 12 than them being with Vgat or Vglut2? Was quality of data not good enough to get that deeper information? If you have used Smartseq, you should have high depth in the sequencing reads to parse out these differences.
8. Very confused by introduction of Switch responsive cells after Voltage-seq results. Are you saying you identified these cells from sequencing experiments?
9. Can switch responses be detected with patch clamp? If you know the spatial identity or location of the neurons, have you tried to replicate this effect?
10. Pnoc is a neuropeptide, I have never read a scientific paper that explicitly states Pnoc is GABA marker. Whereas Gata3 and Nrnx3 are also present in Vglut cells based on your own data.

11. Authors state Switch responses are converging excitatory and inhibitory inputs dominated in a switching manner. Based on voltage-seq, switch response neurons express Tacr1 and Tacr3. Both are excitatory GPCRs that account for excitatory effect on the cells but it is not clear what makes for the inhibitory effect.

Reviewer #2:

None

Reviewer #3:

Remarks to the Author:

In this manuscript, Csillag et al. describe the molecular characterization of neurons classified by all optical electrophysiology recordings using ChR2 and the Voltron voltage indicator. Specifically, they develop a pipeline to stimulate cells expressing ChR2 and record postsynaptic response type from 100s of cells downstream to characterize postsynaptic response types. They repeat this method multiple times to record ~8000 PAG neurons while stimulating VMH terminals. In addition, they build an on-site analysis tool for voltage imaging (Voltview) and use it to guide picking up cells for scRNA seq. To demonstrate the functional properties of their new method, the authors identify sparse GABAergic PAG neurons driven by glutamatergic VMH terminals and use transcriptomics to identify Substance P as a neuromodulator affecting this circuit.

Generally, this is novel and interesting work and develops a method and pipeline for functional connectivity marrying both functional activity and spatial connectomics. This merits publication after addressing some significant and minor points.

Major Points

-It is somewhat unclear from the data presented whether illumination power levels for Voltron with JF-585 will affect ChR2-evoked synaptic release. The authors attempt to address this by showing 3 power levels in extended data fig 2d. However, it is not clear if: 1) Expression level/density of ChR2 will play a role in this, and 2) The authors should repeat this experiment over many neuron pairs and quantify the membrane area of the cells expressing ChR2 exposed to the Voltron imaging light. From a methods usability standpoint: it will significantly add to the paper if the authors are more quantitative in their characterization of cross-activation. Describe instances when the light power needed to image Voltron and/or expression level of ChR2 lead to cross-activation. What is the illumination power threshold? Expression level threshold etc.?

-The authors should discuss how classifying postsynaptic response types differ from readily used calcium imaging with optical stimulation for functional connectivity.

-It will be helpful to compare how Voltseq differs from other spike sorting and voltage imaging analysis that is already published (e.g. Xie et al. Cell Reports, 2021, Adam et al. Nature et al. 2019, etc.). Will Voltseq be available for academic labs to use in their experiments?

Minor Points

-What is the source of the negative deflections in fluorescence of Voltron when shining 473nm light? Is it always there? Does it happen when using other color dyes with Voltron?

-There is no time scale for Fig 1c, extended data fig2. e, g and h.

-For all figures, light powers for stimulation and imaging should be clearly specified in the captions.

-Are there better channelrhodopsin/GEVI pairs that can be used for less crosstalk? Perhaps blue-shifted ChR variants like CheRiff or it's variants? A discussion of this will be useful for the reader.

Author Rebuttal to Initial comments

Response to Referees

Dear Dr. Nina Vogt,

We appreciate your time and effort and the work of the reviewers in providing constructive feedback about our manuscript. In the revised manuscript, we have carefully addressed all the comments and suggestions raised by the reviewers. To answer all the comments, we added new analysis of synaptic drive in the entire VMH-PAG, further clustering analysis of the GABAergic Voltage-Seq RNA-transcriptome, modified and prepared new figure panels. Furthermore, we carried out new experiments to cover the quantitative measurements of all-optical calibration to increase the reproducibility of our work. We describe in detail all the changes in the following point-by-point response. To facilitate the review, we also have highlighted the major edits of the manuscript text in red.

Comments and responses: Reviewer #1:**Remarks to the Author:**

In this work, Csillag et al demonstrate a new methodology to rapidly classify neurons based on their postsynaptic response properties and also to aid transcriptomic interrogation. Authors did commendable effort to achieve this task and provide very impressive datasets to support their claim. They have used soma-targeted Voltron in the connected postsynaptic neurons to achieve all-optical voltage imaging. While this work clearly novel and innovative in combining two different technologies (Voltron and SmartSeq), it is not clear how easily many of the neuroscience labs can utilize the methods that being proposed without spending significant amount of resources and troubleshooting to get this technique established. Though it is clearly step up over patchseq, technical details necessary to reproducibly implement this technique is immensely challenging.

We highly appreciate the effort of the reviewer to contribute with constructive comments as well as valuable suggestions and the overall positive assessment of our manuscript. We attempted to robustly increase the description of details that could help any neuroscience laboratory to set up Voltage-Seq. We believe that the experimental power and cohesiveness of Voltage-Seq is not linearly proportional to the optimization complexity of the workflow.

Major concerns:

1. Voltage imaging identified 18 unique clusters in the PAG based on their response characteristics, but how many of them can be reliably captured with patch clamp electrophysiology? What is the ground truth? Do these clusters exist based on their electrophysiological profile?

We agree with the reviewer that a more detailed discussion of the nature of postsynaptic response types (PRTs) is essential. All-optical voltage imaging captures the physiology of connections, shaped by both the presynaptic and postsynaptic side of the connection. The connection is interrogated by activating the presynaptic side and the postsynaptic response (PR) is captured on the postsynaptic side.

"how many of them can be reliably captured with patch clamp electrophysiology? What is the ground truth?"

We validated the all-optical voltage-imaged PRs with simultaneous whole-cell recording in multiple different cases including recording of subthreshold synaptic excitatory postsynaptic potentials (EPSPs), inhibitory postsynaptic potentials (IPSPs) (Figure 1b), converging synaptic inhibitory and excitatory inputs (Extended Data Figure 2f), the measurements of action potential kinetics (Extended Data Figure 1c-f), estimation of subthreshold detection level (Extended Data Figure 1g), and burst detection optimization (Extended Data Figure 3d). In case of PRs larger

than the Voltron detection sensitivity (~2-2.5 mV), in the whole-cell patch-clamp recordings, we
 have not met any case that would indicate that the imaged PRs would at any level differ from the
 ground truth Patch-clamp electrophysiology.

**“Do these clusters exist based on their electrophysiological profile?”**

We are grateful that the reviewer touched on an important point to clarify. PRTs are
 shaped by both the presynaptic and postsynaptic side of the connection. Amongst other features,
 on the presynaptic side, synaptic strength, release probability, short-term plasticity,
 neurotransmitter modality, and on the postsynaptic side, the number and type of neurotransmitter
 receptors, the intrinsic excitability and firing type of the postsynaptic neuron are all shaping this
 postsynaptic response imaged. When exploring the PRTs within the VMH-PAG connectome, we
 captured the specific PRT composition and distribution, characteristic of the probed VMH-PAG
 pathway. Across the 18 o-phys clusters the strength of the VMH-PAG synaptic drive varies from
 no- or detection-level connection strength (Cluster 18) through only subthreshold response
 (Cluster 17) to various suprathreshold PRTs in the clusters we referred to as “High synaptic drive”
 (Figure 2i, top left). We used these clusters to illustrate how the extracted o-phys parameters
 mapped onto the clusters that defined them (Figure 2i).

In case of weakly firing or purely subthreshold PRTs (Cluster 18,17,16,15), the VMH-PAG
 pathway activation could not reveal the firing type of the postsynaptic PAG neurons. Where the
 synaptic drive was stronger, more suprathreshold activity was observed which for certain clusters
 revealed pronounced intrinsic firing properties (e.g. Cluster 1,5,8). Cluster 1 PAG neurons
 responded with high frequency firing and kept firing after the 1 s blue light train. Cluster 5 PAG
 neurons responded with a separate quick burst upon each activation of VMH terminals at 20Hz
 and immediately stopped after the 1 s blue light train. Cluster 8 PAG neurons displayed 3-5 Hz
 wide bursting at 20 Hz blue light train (Figure 2h).

To attempt to illustrate the relation of synaptic drive and access to firing properties of a
 neuron, we plotted the firing frequency in our VMH-PAG dataset (Response Figure 1a). On the
 tSNE (left) we visualized the firing frequency throughout all the clusters, on the spatial map (right)
 we show that the highest firing frequency in the PRs is not evenly distributed in the PAG, but it
 has higher and lower density areas. To better visualize the spatial pattern of high VMH-PAG
 drive, in (Response Figure 1b), we coronally sliced the 3D map of the whole PAG, and plotted
 the density of only those neurons which had a PR with higher than 40 Hz firing. In (Response
 Figure 1c), we recollected Extended Data Figure 5d to parallel it to our new (Response Figure
 1b) panel. These data demonstrate that PAG areas with the highest level of VMH-PAG activation
 (Response Figure 1b) display PRTs with the most resolved firing-, and intrinsic properties and
 form o-phys clusters further confirmed by their clear spatial segregation into spatial clusters
 (Response Figure 1c). This highlights the tight correlation between the strength of synaptic drive
 and the access to the firing type of the all-optical voltage-imaged neurons. We added these new
 figure panels to Extended Data Figure 4d,e. We discussed this in the result section for Figure 2.

Nevertheless, we performed whole-cell patch-clamp validation of imaged PRs in multiple
 cases. For example, in Extended Data Figure 3f, we did not simultaneously whole-cell record and
 voltage image the shown onset-burst neuron, but first we voltage imaged a field of view (FOV),

we used our on-site analysis, VoltView, to locate a neuron belonging to Cluster 10 with an
 indicated onset bursting PR ("All-optical imaging"). Next, we approached this neuron with the
 patch-pipette and whole-cell-recorded it to test how the optogenetic activation would drive the
 same neuron held at break-in-measured membrane potential ("Opto-e-phys"). Next, we injected
 square pulse currents to characterize the intrinsic firing type of the same neuron ("Current-clamp
 e-phys"). We found an agreement across the lined-up recordings, because the VMH-PAG
 synaptic drive was strong enough to reveal the onset-burst firing of the imaged neuron, which
 was confirmed by the intrinsic current injection-based probing to reproduce this firing behavior.
 Similarly, all-optical voltage imaging revealed the ability of rebound-bursting in "Neuron3" of
 Figure 6g, which we whole-cell patch-clamped and with negative current injections we could
 reproduce the rebound-bursting behavior (Figure 6j).

Taken together, pathway-specific all-optical voltage imaging captures the pathway-
 specific physiology of the interrogated connections; the firing and bursting features captured are
 confirmed by electrophysiology recordings. Importantly, the higher the presynaptic drive, the
 more access we have, to resolve the postsynaptic side, as higher synaptic drive for 1 s more
 closely corresponds to the more constant activation that squared current injections deliver during
 classical whole-cell recording.

2. For spatial mapping of the axons in the PAG, were you using slices (250-300um) that were
 taken for physiology or you imaged thinner slices (50um)?

To estimate the VMH-PAG axonal coverage and design the imaging experiments we used
 separate animals to inject an eYFP-expressing AAV virus into the VMH and to image the axons
 in the PAG (Response Figure 2a, top). We used 50-um-thick sections of the PFA-fixed brains.
 We imaged every third section to cover the PAG with 150 um anterior-
 posterior intervals, because the overall axonal coverage did not vary
 drastically in every 50-100 um, and we were only aiming to have an estimate
 to designate the areas to all-optical image. From the second injection we
 chose identical A-P coordinates. We used the same confocal imaging settings
 for all sections and averaged signal of corresponding coordinates after
 aligning them with custom written MATLAB scripts. We transformed the
 confocal axon images into binary images where every pixel above
 background-level signal was 1, and every pixel with background-level signal
 or below was 0. We mapped these binary images to our 3D map (Response Figure 2, bottom - 6
 example slices mapped) and calculated the axonal density in 3D (Extended Data Figure 3a). The
 3D map was coronally re-sliced to 6 thicker blocks (~600 um each) in Extended Data Figure 3a.
 We added this information to the *Methods* section.

3. For posthoc extraction of the XYZ coordinates from imaged neurons, were the slices cleared?
 What microscope were you using to image these neurons for posthoc spatial analysis?

The reviewer is right that the 3D mapping of the voltage imaged neurons needed a more
 detailed description, thus we revised the *Methods* section "Anatomical 3D mapping":

The videos were taken with a predefined tile positioning. At each A-P coordinate we positioned
 the top left corner of the video on the most dorsal point of the midline in the right hemisphere of
 PAG. The XY table was calibrated so that it moved on the X- and the Y-axis by the length and
 width of videos to capture non-overlapping FOVs. On a hemisphere we used 2 (ML) x 3 (DV)
 FOVs, posterior from Bregma-4.4 we used 3 (ML) x 4 (DV) FOVs as schemed in Figure 2b. To
 extract XYZ coordinates, we first extracted the XY coordinates directly from the videos. We
 measured XY in the center of ROIs relative to the top left corner of the video (Extended

Csillag et. al - Response to Referees

Data Figure 5a, left). The pixel size was 1.25 μm , thus distances in pixels were converted to mm.
 Next, we identified Z coordinates (A-P coordinate) on the imaged side of the brain slice using the
 shape of the Aqueduct which had 50-100 μm A-P accuracy (Extended Data Figure 5b). To build
 Csillag et al. - Extended Data Fig. 5

"guiding rails" for the videos inside the 3D map of the PAG, we created spatial reference lines
 using the SHARP-track package (<https://github.com/cortex-lab/allenCCF>). The package is
 originally used to track electrodes, but it is highly suitable for constructing reference points or
 lines in 3D and extracting the coordinates directly in the CCF4 to support 3D mapping. (Extended
 Data Fig 5a, middle). These lines were predefined coordinates for the predefined FOV positions;
 thus coordinates of a video could be placed on these reference lines by their top left corner at
 any A-P coordinate (Extended Data Fig 5a, right). We added this information to the *Methods*
 section.

 4. How many of the clusters you have seen in Fig. 3 can be also seen with VoltView?

All the o-phys clusters were added to the classifier of VoltView, it can identify any of the 18
 clusters with an accuracy/per clusters shown in Figure 5b, accuracy is approximately 75% in
 average. In the response to the next question, we provide more details on VoltView.

5. Is VoltView a GUI? How user friendly is the software to use? Your current evidence of
 implanting VoltView for sequencing experiments is 60 neurons from 3 wildtype mice. Not sure
 how well this works and can be used by extensive neuroscience labs.

Yes, VoltView has a GUI, we provide a user guide on GitHub, upon favorable assessment of our
 manuscript we deposit the package. The entire voltage imaging analysis from the raw video can
 run on a regular PC in about a minute, it is written to only use CPU, without using parallel
 computation and no GPU thus indeed, no PC with extreme computation capacity is needed for
 the analysis. Here we show some features to illustrate how VoltView works in practice.

- after ROI segmentation VoltView analysis offers the option to have a quick overview and add
 ROIs manually, if necessary, we draw a ROI, and can stop adding ROIs, or add more.

12

Below is an example of how it displays the original FOV with the Voltron-expressing neurons,
 where the color-coded silhouettes represent a suggested cluster for each neuron in the video,
 with the all-optical responses on 7 consecutive sweeps (yellow traces overlaid with the
 subthreshold activity detection) of 24 neurons in each tab. Detected Action potentials are marked
 with magenta points and white thick lines indicate burst activity. Blue lines in the back show the
 timepoints of the optical stimulation analyzed from the video. If the user clicks on a PR (cyan
 rectangle in the top right of each PR) it gets enlarged and the given neuron is indicated on the
 FOV image to locate the neuron of interest. Below we clicked on ROI #23 and ROI #21 and had
 them enlarged, ROI #23 shows delta bursting and o-EPSPs on each sweep upon each optical
 stimulation when not in a burst; ROI #21 is onset bursting and keeps firing o-APs upon each
 optical stimulation. In the bottom of each PR, an average trace is shown, we detect o-EPSPs here
 (can be also done on each sweep separate), note the o-EPSPs (cyan rectangles) in both neurons
 in ROI #23 and ROI #21.

Csillag et. al - Response to Referees

- **FOV Zoom button:** at any View it is going to enlarge the FOV image so that soma positions
 with the ROI ID can be better observed

- **Cluster View button:** switches ROI numbers to cluster IDs, color coded by the cluster ID

- **EQ View button:** changes the default "FLUO View" to the local equalized and contrast
 enhanced Field of View (FOV) image with showing the ROI contours, this shows better the shape
 of the somas before harvesting or patching, as the FLUO view is in 16 bit and brightness may
 differ across ROIs.

In the following example of comparison of 2 videos, **Burst_Freqs_Op** parameter (which is burst
 frequency during the 20 HZ optical stimulation) was user-defined to compare the same neurons
 across the two videos. The 'Cutoff' was ± 20 , thus on the FOV image larger than 120% was
 labeled with red, smaller than 80% was labeled blue, change smaller than \pm cutoff is white. Instead

of re-running the whole analysis for choosing another parameter to compare across the videos,
 the user can choose from the buttons on the bottom left. For any chosen parameter the "cutoff"
 of comparison is the same. In case we would change the 'Cutoff' we can type in a number to the
 "cutoff (%)" edit field and hit Enter. Running the comparison by the same parameter button at
 different "cutoff" levels will show the increase (red) and decrease (blue) with different proportions.
 The first example shows Paired pulse with 10% and 15% cutoff, second is the o-Sub Slope with
 10% and 30% cutoff levels.

Not sure how well this works and can be used by extensive neuroscience labs.

VoltView analysis works the same way on any all-optical voltage imaging video which is
using Voltron-ST-JF585/ChR2(H134). The on-site classification is built on *a priori* data acquired
in the brain structure of interest. First, we generated all-optical voltage imaging data, performed
the clustering analysis, and extracted the cluster centroids. We addressed a simple question, to
find GABAergic neurons, in order to provide proof-of-principle data to support the message
validity of our manuscript. In our case we could have used less *a priori* data to have lower
resolution of classification (less neurons would form less clusters) to locate GABAergic PAG
neurons, but we aimed to give a better description of the PRTs of VMH-PAG. The more imaging
data would be generated and clustered, the better it will define o-phys clusters in a connectome
and have more precise classification using the centroids of these clusters. The classifier of the
VoltView can be re-set for any cell population in any brain region. This is done by a simple
replacement of a table in the VoltView's folder, instructions will be included in the user's guide as
well.

6. Cluster 3,13 and 12, were denser in the posterior PAG in our data (Extended Data Fig. 5b)
and localized in the d-dIPAG. It's not clear what your definition of posterior PAG mean? Bregma-
4.39 onwards?

The reviewer is right, we did not specify the anterior-posterior anatomical ranges of the
VMH-PAG connectome. Based on our VMH-PAG axonal map, we chose to voltage image the
anterior-posterior range between Bregma-3.15 mm and Bregma-5.15 mm. The approximate
middle of this range is around the Lambda (Bregma-4.2 mm). To have a division of anterior vs
posterior VMH-PAG connectome, we used the Lambda as a reference. To make this clearer and
to improve our figures, we added the Lambda coordinate indicator to all the 3D PAG plots in the
manuscript in Figure 3a-e, in Extended Data Figure 6b,d and in Figure 6a.

7. When Voltage-seq was implemented did you discover what is unique about Cluster 3,13 and
12 than them being with Vgat or Vglut2? Was quality of data not good enough to get that
deeper information? If you have used Smartseq, you should have high depth in the sequencing
reads to parse out these differences.

The reviewer had a valid question as we could have attempted to further dissect our RNA-
transcriptome sample. The quality of RNA-sequencing of single neurons that had been voltage-
imaged before is highly satisfactory, the number of genes detected per sample was ~6000
genes/sample, this is the quality of Smart-Seq2 data mostly because of its robust library prep
chemistry and extensive read coverage. At the same time, the inhibitory diversity is extremely
high in the PAG also reported by a robust recent work from the Dulac laboratory at the Allen
Institute (we cite this work
below as well in *response 10*),
which single-nucleus RNA-
sequenced ~100 000 PAG
neurons and distinguished 63
excitatory and 57 inhibitory
neuronal clusters. That also
implies that for a thorough
description and molecular
classification of GABAergic
neurons it would be necessary
to work with a much higher number of samples in the order of many hundreds. This estimation is
based on our experience in Patch-Seq experiments where exclusive molecular markers are
discovered with at least 15-20 samples of the same firing type.

However, the question of the reviewer made us curious whether further clustering our
 GABAergic neurons would form subclusters and if so, whether we could identify genes which are
 enriched in the new subclusters. We performed the new analysis using only the GABAergic VMH-
 PAG neurons and we managed to reveal a
 polarization of GABAergic molecular
 identity. Despite this moderate separation,
 the agreement of the two subclusters to the
 PRT clusters showed a trend but not a
 clear correlation. In our VS-Cluster2 we
 had proportionally higher number of
 neurons belonging to o-phys Cluster 3 than
 to VS-Cluster1, but we would be extremely
 cautious with considering this trend a solid
 base. Also, it is important to keep in mind that we had some of the VoltView-suggested neurons
 which turned out to be Glutamatergic. This is expected, because rebound firing - characteristic
 of o-phys Cluster 3 - even within the VMH-PAG connectome can occur on a multitude of cell
 types, and as a PRT is not equivalent of the firing type, such purely biological questions of identity
 must be carefully addressed with an increased number (preferably hundreds) of samples, which
 is beyond the scope of our current methodical work.

Using our new analysis, we could identify molecular markers that were enriched in VS-
 Cluster1, for example *Scp2*, *Atp5*, *Smim19* or *Rps5* (first row of Figure) or in VS-Cluster2, like
 *Grin2b*, *Kcnq1ot1*, *Meg3*, or *Trpm7*. In the bottom row, we plotted control genes (*Tuba1a*, *Snap25*
 as general neuronal genes and *Gad1*, *Gad2* for GABAergic identity) to visualize that any
 differences above are not due to overall sample quality differences. Though exclusive markers
 are excessively challenging to be identified in a small sample from a highly diverse population,
 we could identify genes which were enriched in our further divided GABAergic subclusters.

Taken together, the Voltage-Seq RNA-transcriptome quality is sufficient to gain an insight
 to the gene expression profile of all-optical voltage imaged neurons with the cohesive addition to
 explore the correlations of neuronal identity and PRTs within a given connectome.

8. Very confused by introduction of Switch responsive cells after Voltage-seq results. Are you
 saying you identified these cells from sequencing experiments?

The reviewer is right when pointing out that the introduction of the Switch responder PAG
 neurons was confusing in the end of the section belonging to Figure 5, which described the
 workflow of Voltage-Seq. As we present proof-of-principle experiments on the closer investigation
 of PAG neurons with Switch response using our combined approaches in the section belonging
 to Figure 6., we restructured Figure 5. and Extended Data Figure 7., and edited the corresponding
 texts accordingly. We focused all the text and figures describing the Switch disinhibitory motif into
 the last section which made the manuscript structure clearer.

 9. Can switch responses be detected with patch clamp? If you know the spatial identity or
 location of the neurons, have you tried to replicate this effect?

We attempted to test the switch behavior multiple times in whole-cell recordings performed dafter
 all-optical voltage imaging, when we characterized the Switch responders in (Figure 6, Response
 Fig 5a). In whole-cell current-clamp with 0 pA current injected (in this configuration we did not
 interfere with the membrane potential of the neuron, only recorded it) we used 3 trains of 20 Hz

optogenetic activation of the VMH-PAG terminals for 1s each. We could not observe the switch
 behavior, but we were able to observe both IPSPs and APs (Response Fig 5b). We validated the
 presence of monosynaptic excitation at -70 mV (VMH input) and disinhibitory inhibition at +10 mV
 (local PAG GABAergic input) in voltage-clamp of the same neuron (Figure 5d, Response Fig 5c).
 We hypothesized that the perturbation of the intracellular ionic milieu caused by the cell dialysis
 during whole-cell recording may have disrupted the mechanism of switching. Our hypothesis is
 that Switch responders are bistable, and they can maintain prolonged depolarization after being
 excited. We did not dissect the ionic mechanisms of the Switch responders, what we may explore
 in the future, but we did observe prolonged depolarization on some of these neurons.

Interestingly, if we recorded the neuron closer in time to the break-in to whole-cell configuration,
we could observe some maintained or unstable depolarization (green, Response Fig 5d), but if
we recorded the same neuron ~10 min later, these depolarizations were not present (Response
Fig 5e). This observation further supports our hypothesis that whole-cell dialysis may be the
cause of the disruption of the switch behavior.

10. *Pnoc* is a neuropeptide, I have never read a scientific paper that explicitly states *Pnoc* is
GABA marker. Whereas *Gata3* and *Nrxn3* are also present in Vglut cells based on your own data.

We successfully attempted to find GABAergic neurons solely based on their activity and
their PRT classification within the VMH-PAG connectome. We designed the sample size of
Voltage-Seq to be able to achieve this goal and extracted putative marker genes which are
enriched in the "Excitatory" or "Inhibitory" clusters which the unbiased clustering segregated. High
diversity is expected within both clusters. Here we show result from a robust work from the Dulac
Lab (Vaughn et al.). In this work, large dataset (snRNA-Seq on ~100,000 PAG cells) clustering
formed large number of clusters (63 excitatory, 57 inhibitory). Cluster analysis for marker search
would normally implement a cutoff to exclude genes which are not expressed in at least 30-40%
of the neurons of the investigated cluster. Vaughn et al. filtered out differentially expressed genes
expressed in less than 40% of neurons of a cluster to focus on transcripts that can best delineate
clusters. Because of the high diversity within both our two clusters, with this approach, we would
have distorted our data and potentially useful information could have been lost (e.g., *Tacr1* or
*Tacr3* expressed in 12 neurons). We attempted to use our data as unprocessed as possible and
showed it unfiltered. However, as quality-proof, we could use our Voltage-Seq data beyond the
separation of "Excitatory" and "Inhibitory" neurons (as shown in response 7.) and found genes
enriched in the "Inhibitory" PAG neurons such as *Pnoc*, *Gata3* or *Nrxn3*. Both *Gata3* and *Nrxn3*
are expressed in 4-5 of the excitatory neurons with a median of 0 and 0 TPM respectively, while
both genes are expressed in almost all the inhibitory neurons with a median of 5.9 and 7.6 TPM
respectively (Figure 6f). Filtering of genes expressed in less than 40% in this case would have
made both markers 100% exclusive, however we only considered these as highly enriched
GABAergic markers. The corresponding Allen database ISH of these genes is available in
coronal sections, that allowed a validating indication of the spatial pattern of these enriched genes
by a comparison to the spatial distribution of GABAergic PAG neurons (Extended Data Figure
8c).

The reviewer pointed to one of the scientific novelties our manuscript revealed to provide
sufficient proof-of-principle data about VMH-PAG connectome, including molecular properties of

GABA neurons or circuit motifs like the Switch motif. Here we show a summary of neuropeptide-
expressing PAG clusters from the work of Vaughn et al., analyzed based on whether certain
neuropeptides define excitatory or rather inhibitory clusters. In agreement with our Voltage-Seq
dataset (Figure 6f), *Pnoc* gene encoding Prepronociceptin defined the highest number of
GABAergic inhibitory clusters compared to all the neuropeptide genes detected in the PAG (Fig6th
gene from the left). Taken together, our proof-of-principle Voltage-Seq dataset could capture

novel robust properties of GABAergic neurons, though our goal was simpler, to find GABAergic
 neurons in the VMH-PAG connectome.

11. Authors state Switch responses are converging excitatory and inhibitory inputs dominated in
 a switching manner. Based on voltage-seq, switch response neurons express Tacr1 and Tacr3.
 Both are excitatory GPCRs that account for excitatory effect on the cells but it is not clear what
 makes for the inhibitory effect.

The alternate excitatory and inhibitory behavior of the Switch motif is driven by ionotropic
 synaptic excitatory (Glutamate) and inhibitory (GABA) inputs resulting in fast excitatory
 postsynaptic responses (EPSPs) and inhibitory postsynaptic responses (IPSPs) detected in the
 voltage imaging as o-EPSPs or o-IPSPs (Figure a,b). Beyond this being confirmed by the
 temporal kinetics of the imaged responses, we had tested the ionotropic nature with Gabazine,
 selective GABAA antagonist, which blocked the GABAergic transmission, and all the responses
 of a Switch responder became exclusively excitatory (Figure c). We could not block the
 glutamatergic transmission because in the all-optical stimulation we would have eliminated all the
 excitatory VMH-PAG synaptic drive. The Switch behavior is driven by fast synaptic responses,
 neuromodulators take their effect in many seconds to minutes.

If we ought to hypothesize a possible mechanism responsible for the switch behavior, it
 may be the bistability of the GABAergic neurons displaying a switch. Bistable neurons change
 their resting potential to two or multiple different membrane potentials, in other words these
 neurons have multiple "resting" potentials of which one could be suprathreshold (Fuentelba et
 al.; Lee and Heckman; Royer, Martina and Pare). As a result, neurons would stay depolarized
 after depolarization, this is when inhibition is observed as the neuron is further from its GABA
 reversal potential; and the neurons stay hyperpolarized after receiving the inhibition, this is when
 only excitation is observed, as the neuron is very close to its GABA reversal potential, where
 GABA has no ionic driving force. These inputs are coming from different sources as we could
 separate monosynaptic excitation from the VMH-PAG and disynaptic inhibition from the local
 PAG inhibitory circuitry (Figure 6d).

Before we performed Substance P (SP) pharmacology on Switch motifs, we also carried
 out patch-clamp recordings on PAG neurons during exposure to SP to measure the time window
 of the SP-induced depolarization. This was necessary to know, so that we could run all-optical
 voltage imaging when the pharmacological effect could already show its effect. SP-induced
 depolarization needs many seconds, even tens of seconds to reach maximal effect, thus it could
 not create EPSPs.

In summary, based on our Voltage-Seq RNA-transcriptome data we could identify
 neuromodulator receptors on inhibitory PAG neurons, of which we presented SP receptors as an
 example. We could successfully validate the SP neuromodulatory effect on the Switch responder
 neurons in a living PAG circuitry. Our experiment also revealed that SP has a depolarizing effect
 on these neurons, which had not been reported before.

**Comments and responses: Reviewer #2:**

None

**Comments and responses: Reviewer #3:**

Remarks to the Author:

In this manuscript, Csillag et al. describe the molecular characterization of neurons classified by
all optical electrophysiology recordings using ChR2 and the Voltron voltage indicator.399 Specifically, they develop a pipeline to stimulate cells expressing ChR2 and record postsynaptic
response type from 100s of cells downstream to characterize postsynaptic response types.401 They repeat this method multiple times to record ~8000 PAG neurons while stimulating VMH
terminals. In addition, they build an on-site analysis tool for voltage imaging (Voltview) and use
it to guide picking up cells for scRNA seq. To demonstrate the functional properties of their new
method, the authors identify sparse GABAergic PAG neurons driven by glutamatergic VMH
terminals and use transcriptomics to identify Substance P as a neuromodulator affecting this
circuit.

Generally, this is novel and interesting work and develops a method and pipeline for functional
connectivity marrying both functional activity and spatial connectomics. This merits publication
after addressing some significant and minor points.

We highly appreciate the constructive comments of the reviewer as well as the suggestions to
improve reproducibility and the overall favorable assessment of our manuscript.

**Major Points**416 1. It is somewhat unclear from the data presented whether illumination power levels for Voltron
with JF-585 will affect ChR2-evoked synaptic release. The authors attempt to address this by
showing 3 power levels in extended data fig 2d. However, it is not clear if: 1) Expression
level/density of ChR2 will play a role in this, and 2) The authors should repeat this experiment
over many neuron pairs and quantify the membrane area of the cells expressing ChR2 exposed
to the Voltron imaging light. From a methods usability standpoint: it will significantly add to the
paper if the authors are more quantitative in their characterization of cross-activation.423 The reviewer had an important addition that will increase the reproducibility of the methodology.
We carried out new experiments to increase our data about the possible crosstalk of the
ChR2(H134)/Voltron pair we used for all-optical imaging. We whole-cell recorded VMH neurons
that expressed the ChR2 for 4-5 weeks. We chose this relatively long expression time because
the Voltron sensor needs 4-5 weeks to express at the high levels as it can only localize in the
neuronal membrane, thus in comparison with cytosolic Ca-indicators the available surface is
extremely limited. We inject the ChR2-expressing virus to the VMH and the Voltron-ST-
expressing virus to the PAG during the same transcranial surgery, (we have not indicated this in
the manuscript, but we added this information to the *Materials and Methods* section). Parallel
with the Voltron-ST expression, ChR2 expression has the same time window of 4-5 weeks. Thus,
we used identical conditions to our experiments presented in the manuscript and this long
expression time of the ChR2 resulted in a high expression level of the opsin in the VMH.435 We first validated the ChR2 expression by whole-cell recording of the neurons and with the
application of 2 ms, 2.5 mW/mm², 473 nm blue light pulses, where in every neuron we evoked
firing of single or multiple APs by each blue light stimulation. We added biocytin to the patch
pipette solution and filled the recorded VMH neurons. Upon the immunohistochemical

visualization of biocytin, we acquired Z-stacked confocal microscope images of the recorded
 neurons to attempt to find a correlation between the expression level of ChR2 and the measured
 ChR2-mediated response amplitude. We included here an example to illustrate that ChR2 is

irregularly dot-patterned and is not restricted to the soma, it is present on the proximal and distal
 dendrites as well. Besides, local axons and synapses terminating on the recorded neuron are
 difficult to tell apart from the ChR2 expressed by the neuron. As the visual quantification of ChR2
 proved to be excessively challenging, we chose to rely on measuring the ChR2-mediated
 currents in whole-cell recordings to test for possible crosstalk of ChR2 and the 585 nm imaging
 light. Recordings were carried out in 4 animals, all of them with the same 4-5 weeks of ChR2
 expression time. When measuring the ChR2 activation by 585 nm light we compared the original
 imaging 585 nm illumination light power (~ 14 mW/mm²), half and double of that (~ 7 mW/mm²
 and ~ 28 mW/mm²). We found that with our original example in Extended Data Figure 2c, we

presented a case of cross-activation somewhat above the average. We measured 1.12 ± 1.1 mV
 with ~ 28 mW/mm², 0.71 ± 0.5 mV with ~ 14 mW/mm² and 0.31 ± 0.2 mV with ~ 7 mW/mm² 585
 453 nm illumination power ($n = 9$). Based on our new data we could conclude that the possible 585
 454 nm illumination-originated depolarization of synaptic terminals during all-optical experiments
 could be approximately up to ~ 1 mV. To further validate the lack of cross activation of synapses,
 we whole-cell recorded PAG neurons and used the same light power intensities to attempt to
 evoke synaptic release. First, we validated the connectivity on our recorded PAG neurons by the
 2.5 mW/mm² 473 nm blue light pulses, and then we deployed the 585 nm yellow light pulses.
 We were not able to evoke any synaptic release by the 585 nm illumination at any levels including
 double the imaging light power of ~ 28 mW/mm². We prepared new figures of the recorded data
 and included them to the modified Extended Data Figure 2a-c.

2. Describe instances when the light power needed to image Voltron and/or expression level of
 ChR2 lead to cross-activation. What is the illumination power threshold? Expression level
 threshold etc.?

The reviewer had an important suggestion, we could include information about an
 extreme case of ChR2 cross-activation. With our optical path used and with our light source
 (Spectra X, Lumencore) we were not able to induce substantial (>2-3mV) ChR2 cross-activation.
 Nevertheless, during the optimization of the light path in the first pilot experiments we tested
 another dual-band excitation filter as well, the Chroma 59022x, which had the yellow light
 reflected between 550-590 nm. 100% illumination with our light source through this excitation
 filter (~28 mW/mm²) activated the somatic ChR2 resulting in ~5 mV ChR2-signals, and on 50%
 illumination (what we used for the imaging) induced ~2-3 mV ChR2-response (~14 mW/mm²). In
 our experiments this would mean that we would depolarize the VMH-PAG synapses by 2-3 mV,

which would not be a robust effect. However, we used a more strict dual-band excitation filter to
 acquire the data of the manuscript and the new calibration data, the Chroma 59010x, that
 reflected the yellow light between 570-595 nm and as we showed in Extended Data Figure 2., it
 resulted in an extreme maximal 2 mV, but in average 0.71 ± 0.5 mV yellow-light-induced ChR2-
 response at 100% illumination (~28 mW/mm²), and where we used it for the experiments with
 50% illumination (~14 mW/mm²) we induced ~1 mV or less but in average 0.31 ± 0.2 mV
 depolarization. The yellow light is not applied in a train but in a single pulse to illuminate the FOV,
 thus it would depolarize - in our case - the synapses by ~1 mV, which is not a substantial
 interference with the resting potential of boutons which in the literature of resting potential of a
 synapse is estimated to be around -50-60 mV as an example of whole-bouton patch clamping
 shows the voltage-current relations of a synapse (Novak et al.).

 3. The authors should discuss how classifying postsynaptic response types differ from readily
 used calcium imaging with optical stimulation for functional connectivity.

The reviewer is right, a short discussion of the comparison of all-optical circuit interrogation with
 Ca-imaging and voltage imaging would be beneficial for the manuscript. We updated the
 discussion with the following: "In functional connectivity studies calcium imaging is suitable to
 monitor robust suprathreshold activity on multiple neurons simultaneously. However, even the
 latest GCaMP calcium indicators (Zhang et al. 2023) have low temporal resolution to resolve
 single APs with high fidelity and low sensitivity to report subthreshold activity. Before the
 appearance of GEVIs, inhibition was a blind spot for population imaging. Voltage imaging gives
 access to subthreshold events of both polarities and has the temporal resolution necessary to
 interrogate PRTs."

4. It will be helpful to compare how Voltseq differs from other spike sorting and voltage imaging
 analysis that is already published (e.g. Xie et al. Cell Reports, 2021, Adam et al. Nature et al.
 2019, etc.). Will Voltseq be available for academic labs to use in their experiments?

VoltView is analyzing all-optical Voltron imaging by default, it is built in a way that it is looking for
 the blue light artifacts delivered by the blue optogenetic stimulation and reported by the FRET
 Voltron sensor (see some more description below in "Minor comments"). Furthermore, VoltView

does not contain a self-built segmentation algorithm which differs across all available analysis
 packages, *VoltView* has *Cellpose* built-in, which is a very highly trained and well-optimized
 generalist segmentation package. A big advantage of *Cellpose* that it recognizes the somas and
 only segments the somatic pixels instead of placing out a rectangle around the soma which would
 include background or dendrites of other neurons. Our *Concentric* analysis defines the “Inner”
 and “Outer” areas to clean the data from single photon background and can do this because of
 the precise segmentation with *Cellpose*. Most importantly, *VoltView* is not only filtering out traces
 that we can visually further explore, but detects APs, EPSPs, IPSPs, Bursts and extracts the 29
 different o-phys parameters for classification of PRTs. Also, *VoltView* has a built-in classifier
 which can be updated even after each experiment, so that the classification of PRTs will get more
 and more accurate by involving more and more o-phys data into the clustering analysis that
 generates the cluster centroids what *VoltView* is using for the on-site classification.
 Moreover, multiple video comparison mode has interactive settings on-site to switch between
 parameters to define the color coding of increase or decrease of the chosen parameter in each
 imaged neuron (e.g. burst length in Figure 4). We added this information into the Methods section.
 *VoltView* will be available on GitHub upon acceptance of our manuscript.

Minor Points

1. What is the source of the negative deflections in fluorescence of Voltron when shining 473nm
 light? Is it always there? Does it happen when using other color dyes with Voltron?

The reviewer pointed out an interesting feature of the Voltron sensor, indeed, we did not discuss
 the origin of the blue light artifacts in detail. The Voltron sensor is a FRET protein which during
 imaging is continuously charged by a flow of power delivered by the imaging yellow (585 nm)
 light. The power of the blue (473 nm) light when simultaneously applied with the yellow light adds
 up to it, thus we see the summation of the two light powers. This added energy results in a positive
 increase in the fluorescence, the Voltron becomes brighter. The Voltron reports positive voltage
 deflection with dimming and negative voltage deflection with increased brightness. This means
 that blue light artifacts, or “Blue spikes” as we call them, always have reversed polarity to action
 potentials thus they cannot be mixed up. The Blue spike is only detected by the Voltron sensor,
 it does not influence the membrane potential of the imaged neuron, as we can appreciate this on
 (Figure 1b) simultaneous whole-cell recording and voltage
 imaging of inhibitory postsynaptic potentials. There was no artifact
 removed from the black electrophysiology trace. We find this
 property of the Voltron useful as it provides a confirming
 timestamp to each optical stimulation.

We could not test whether Blue spikes would appear with other
 JF-dyes, because our optical setup has the double-band filter
 customized for JF-585. At the same time, to avoid crosstalk of the
 optogenetic stimulation and voltage imaging we would not use
 any other dyes with shorter wavelengths and would not
 recommend the JF-635 either as the dF/F flexibility of the Voltron
 with that Halo-Tag conjugate is much lower than with all the other dyes (JF-585, JF-549, JF- 525,
 JF-503) (Abdelfattah et al.). Besides, we would hypothesize that the FRET sensor would detect
 any other light as its energy would be added to the used illuminating light.

2. There is no time scale for Fig 1c, extended data fig2. e, g and h.

We thank the reviewer for pointing out the absence of time scales in the mentioned panels, we
 inserted those and checked through all the other panels as well.

3. For all figures, light powers for stimulation and imaging should be clearly specified in the
captions.

We thank the reviewer for pointing out the absence of light power in the captions, we inserted
those to the main and extended data figures as well.

4. Are there better channelrhodopsin/GEVI pairs that can be used for less crosstalk? Perhaps
blue-shifted ChR variants like CheRiff or it's variants? A discussion of this will be useful for the
reader.

We would not particularly suggest that a better combination of opsin/GEVI is needed, because
based on our original and newly recorded data the approximately estimated ~0.2-0.5 mV
depolarization of synaptic terminals is a negligibly small crosstalk which is unlikely to significantly
influence the all-optical experiments. We agree with the reviewer that other opsin/GEVI pairs are
important to be referenced, we highlighted the combination of one of the first opsin/GEVI pairs
used for all-optical voltage imaging, the CheRiff/ Quasar2 (Fan et al.) pair used by the Cohen lab.

Overall, we want to thank the reviewer for assessing our work and we are grateful to for the
constructive comments. We were glad to read that the reviewer found our work novel and
interesting. We also would like to add that we firmly believe the experimental power and
cohesiveness of Voltage-Seq is not linearly proportional to the optimization complexity of the
workflow. We added more details and new calibration data to the description of the methodology
in every level in our revised manuscript.

References

Abdelfattah, A. S., et al. "Bright and Photostable Chemigenetic Indicators for Extended in Vivo Voltage
Imaging." *Science* 365.6454 (2019): 699-704. Print.

Fan, L. Z., et al. "All-Optical Physiology Resolves a Synaptic Basis for Behavioral Timescale Plasticity."
*Cell* 186.3 (2023): 543-59.e19. Print.

Fuentealba, P., et al. "Membrane Bistability in Thalamic Reticular Neurons During Spindle Oscillations." *J*
*Neurophysiol* 93.1 (2005): 294-304. Print.

Lee, R. H., and C. J. Heckman. "Bistability in Spinal Motoneurons in Vivo: Systematic Variations in
Rhythmic Firing Patterns." *J Neurophysiol* 80.2 (1998): 572-82. Print.

Novak, P., et al. "Nanoscale-Targeted Patch-Clamp Recordings of Functional Presynaptic Ion Channels."
*Neuron* 79.6 (2013): 1067-77. Print.

Royer, S., M. Martina, and D. Pare. "Bistable Behavior of Inhibitory Neurons Controlling Impulse Traffic
through the Amygdala: Role of a Slowly Deactivating K⁺ Current." *J Neurosci* 20.24 (2000): 9034-
9. Print.

Vaughn, Eric, et al. "Three-Dimensional Interrogation of Cell Types and Instinctive Behavior in the
Periaqueductal Gray." *bioRxiv* (2022): 2022.06.27.497769. Print.

Decision Letter, first revision:

Dear Dr. Fuzik,

Thank you for submitting your revised manuscript "Voltage-Seq: all-optical postsynaptic connectome-guided single-cell transcriptomics" (NMEMTH-A51098B). It has now been seen by the original referees and their comments are below. The reviewers find that the paper has improved in revision, and therefore we'll be happy in principle to publish it in Nature Methods, pending minor revisions to satisfy the referees' final requests and to comply with our editorial and formatting guidelines.

TRANSPARENT PEER REVIEW

Nature Methods offers a transparent peer review option for new original research manuscripts submitted from 17th February 2021. We encourage increased transparency in peer review by publishing the reviewer comments, author rebuttal letters and editorial decision letters if the authors agree. Such peer review material is made available as a supplementary peer review file. Please state in the cover letter 'I wish to participate in transparent peer review' if you want to opt in, or 'I do not wish to participate in transparent peer review' if you don't. Failure to state your preference will result in delays in accepting your manuscript for publication.

ORCID

Best regards,
Nina

Nina Vogt, PhD
Senior Editor
Nature Methods

Reviewer #1 (Remarks to the Author):

All my concerns are addressed. This manuscript is suitable for publication.

Reviewer #3 (Remarks to the Author):

I thank the reviewers for addressing both the major and minor points. The authors have now added more data to characterize and guide reproducibility of the all optical imaging part of the manuscript. I have no further major concerns about the manuscript.

minor comments:

1. I recommend that the data included in response to major point 2 be included in the methods/supplement. A detailed description similar to the one in the reviewer response is appropriate. Showing an extreme case of cross talk and the process of trying different cubes and optimizing the light path is informative for the reader.
2. Caption of fig. 1 has a missing standard deviation value in this sentence: 'We validated that ~14 mW JF-585 excitation had a negligible $\sim 0.7 \pm X$ mV...

Final Decision Letter:

Dear Janos,

I am pleased to inform you that your Article, "Voltage-Seq: all-optical postsynaptic connectome-guided single-cell transcriptomics", has now been accepted for publication in Nature Methods. Your paper is tentatively scheduled for publication in our September print issue, and will be published online prior to that. The received and accepted dates will be December 14th, 2022 and June 21st, 2023. This note is intended to let you know what to expect from us over the next month or so, and to let you know where to address any further questions.

Once your paper is typeset, you will receive an email with a link to choose the appropriate publishing options for your paper and our Author Services team will be in touch regarding any additional information that may be required.

Please note that *Nature Methods* is a Transformative Journal (TJ). Authors may publish their research with us through the traditional subscription access route or make their paper immediately open access through payment of an article-processing charge (APC). Authors will not be required to make a final decision about access to their article until it has been accepted. [Find out more about Transformative Journals](https://www.springernature.com/gp/open-research/transformative-journals)

Your paper will now be copyedited to ensure that it conforms to Nature Methods style. Once proofs are generated, they will be sent to you electronically and you will be asked to send a corrected version within 24 hours. It is extremely important that you let us know now whether you will be difficult to contact over the next month. If this is the case, we ask that you send us the contact information (email, phone and fax) of someone who will be able to check the proofs and deal with any last-minute problems.

If, when you receive your proof, you cannot meet the deadline, please inform us at rjsproduction@springernature.com immediately.

Once your manuscript is typeset and you have completed the appropriate grant of rights, you will receive a link to your electronic proof via email with a request to make any corrections within 48 hours. If, when you receive your proof, you cannot meet this deadline, please inform us at rjsproduction@springernature.com immediately.

Once your paper has been scheduled for online publication, the Nature press office will be in touch to confirm the details.

Once your paper has been scheduled for online publication, the Nature press office will be in touch to confirm the details.

Content is published online weekly on Mondays and Thursdays, and the embargo is set at 16:00 London time (GMT)/11:00 am US Eastern time (EST) on the day of publication. If you need to know the exact publication date or when the news embargo will be lifted, please contact our press office after you have submitted your proof corrections. Now is the time to inform your Public Relations or Press Office about your paper, as they might be interested in promoting its publication. This will allow them time to prepare an accurate and satisfactory press release. Include your manuscript tracking number NMETH-A51098C and the name of the journal, which they will need when they contact our office.

About one week before your paper is published online, we shall be distributing a press release to news organizations worldwide, which may include details of your work. We are happy for your institution or funding agency to prepare its own press release, but it must mention the embargo date and Nature Methods. Our Press Office will contact you closer to the time of publication, but if you or your Press Office have any inquiries in the meantime, please contact press@nature.com.

Nature Portfolio journals [encourage authors to share their step-by-step experimental protocols](https://www.nature.com/nature-research/editorial-policies/reporting-standards#protocols) on a protocol sharing platform of their choice. Nature Portfolio 's Protocol Exchange is a free-to-use and open resource for protocols; protocols deposited in Protocol Exchange are citable and can be linked from the published article. More details can found at www.nature.com/protocolexchange/about.

Please note that you and any of your coauthors will be able to order reprints and single copies of the issue containing your article through Nature Portfolio 's reprint website, which is located at <http://www.nature.com/reprints/author-reprints.html>. If there are any questions about reprints please send an email to author-reprints@nature.com and someone will assist you.

Best regards,
Nina

Nina Vogt, PhD
Senior Editor
Nature Methods